

# Quantum criticality in many-body parafermion chains

**Ville Lahtinen[1], Teresia Mansson[2] and Eddy Ardonne[3]⋆**

**1** Dahlem Center for Complex Quantum Systems,
Freie Universität Berlin, 14195 Berlin, Germany
**2** Department of Theoretical Physics, School of Engineering Sciences,
Royal Institute of Technology (KTH), Roslagstullsbacken 21, SE-106 91 Stockholm, Sweden
**3** Department of Physics, Stockholm University, AlbaNova University Center,
SE-106 91 Stockholm, Sweden

⋆ ardonne@fysik.su.se

## Abstract

We construct local generalizations of 3-state Potts models with exotic critical points. We analytically show that these are described by non-diagonal modular invariant partition functions of products of $Z_3$ parafermion or $u(1)_6$ conformal field theories (CFTs). These correspond either to non-trivial permutation invariants or block diagonal invariants, that one can understand in terms of anyon condensation. In terms of lattice parafermion operators, the constructed models correspond to parafermion chains with many-body terms. Our construction is based on how the partition function of a CFT depends on symmetry sectors and boundary conditions. This enables to write the partition function corresponding to one modular invariant as a linear combination of another over different sectors and boundary conditions, which translates to a general recipe how to write down a microscopic model, tuned to criticality. We show that the scheme can also be extended to construct critical generalizations of $k$-state clock type models.

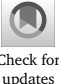

# 1 Introduction

At a quantum critical point two distinct phases of matter coexist. A remarkable feature of 1D systems is that such special points in the phase diagram are in general described by a field theory with conformal symmetry – a conformal field theory (CFT) [1, 2]. In other words, the system exhibits a universal behavior regardless of the underlying microscopic model, i.e. what are the local degrees of freedom and how they interact. This universal description at the critical point enables to determine what phases of matter appear in the vicinity of the critical point when the system is perturbed away from it. However, the relation of the universality class to criticality of a microscopic model is a one-way problem. If a given microscopic model exhibits a critical point with diverging correlation length, given a catalog of possible CFTs it is comparatively easy to make ansatzes to verify which CFT describes it. The converse is not true though. While symmetries present in the CFTs constrain the possible microscopic models, there is in general no recipe to write down a model that exhibits a critical point described by a given CFT.

The lack of framework to write down a microscopic model for a given CFT is an outstanding problem from several perspectives. From an academic perspective, it is of interest to understand the minimal microscopic conditions that can give rise to a given universal behavior. This enables to experimentally search for such behavior in existing or synthesized materials. While CFT is a standard theoretical tool, experiments probing CFT predictions beyond measuring critical exponents are still few [3,4]. Due to the coexistence of phases of matter at the critical points, critical models with universal behavior that are perturbed away from the critical point can also serve as starting points for models of gapped, possibly topological phases of matter. Recently, the interplay of symmetry-protected topological order and quantum criticality has been explored in spin-1/2 chains [5–7], with a unified picture emerging how the protecting symmetries and the possible fractionalized edge states determine the universality classes of transitions [8,9].

While the universal description of a critical point is a property of 1D systems (the nature of conformal critical points in 2D is an open question and a subject of cutting-edge numerical studies [10, 11]), it has been proposed that critical 1D systems can serve as building blocks

of 2D topologically ordered states of matter [12–15]. Critical 1D systems can be arranged into 2D array and when coupled together in a designed fashion, the system can become a gapped topologically ordered state whose nature depends only on the couplings and on the CFT describing the critical 1D systems. Thus new microscopic models with exotic critical points enable also the construction of new models for exotic 2D topological states of matter. In particular, since the advent of topological quantum computation, there is much interest to construct states that harbour non-Abelian anyons that could be employed for topologically protected quantum information processing [16].

Motivated by these open questions, in this work we advance the program initiated in our earlier works, where we constructed exactly solvable spin-1/2 chains for all criticalities in the $so(N)_1$ universality class [6, 17, 18]. Instead of spin chains, we focus here on 3-state Potts models, where the local degrees of freedom are not spins, but clock variables, and construct generalizations that exhibit critical points whose critical behavior has not been previously discussed in the literature. 3-state Potts models themselves have recently attracted attention due to their relation to parafermion modes [19], that have been proposed to be realized, following the same principles that lead to the recent experimental discovery of Majorana modes [20,21], by inducing superconductivity via the proximity effect on fractional quantum Hall edge states, such as the $\nu = 1/3$ Laughlin state [22, 23]. A uniform array of such parafermion modes is unitarily equivalent to a 3-state Potts model via the parafermion version of the Jordan-Wigner transformation, the Fradkin-Kadanoff transformation [24]. The recent focus on parafermions arises not from themselves though, but from their collective behavior. Were they to hybridize in a 2D array, they could realize a state that hosts the coveted Fibonacci anyons that are universal for quantum computation [14]. This prospect, while speculative, has motivated research into 1D collective states of parafermion modes. Different parafermion phases have been understood from the symmetry protection perspective [25], the microscopics of $Z_3$ parafermion CFT describing the criticality of the 3-state Potts model has been analyzed [26] and the phase diagram in the presence of longer range parafermion tunneling has been studied [27].

While our primary motivation is the construction of new Potts-like models with exotic criticalities, our models turn out to be unitarily equivalent to parafermion chains with many-body interactions between the parafermion modes. Thus as a by-product of our construction we address the nature of the critical behavior of parafermion chains in the presence of such many-body terms that may also arise when parafermion modes hybridize. To our understanding, the effect of such many-body terms has not been considered previously in literature. We find that when many-body terms are comparable to local chemical potential-like terms, the chains are critical and described by the non-diagonal modular invariant partition functions of a CFT that is product of some number of $Z_3$ parafermion or $u(1)_6$ CFTs.

Usually, when talking about a critical point being described by a given CFT, one refers to the diagonal invariant. Our work explicitly shows that this is too restrictive. We demonstrate that a particular class of non-diagonal modular invariant partition functions – the permutation invariants – that are usually overlooked when talking about physical systems, also have local microscopic models that realize them. Before proceeding to the actual constructions and the physics underlying them, we summarize our main results – the microscopic generalizations of the 3-state Potts models and the exotic critical points they exhibit.

## 1.1 Summary of results: Generalized critical 3-state Potts models

The starting point for our construction is the critical nearest neighbour 3-state Potts model that is described by the Hamiltonian (see for instance Ref. [28])

$$H_\pm = \pm \sum_{i=1}^{L} \left( X_i X_{i+1}^\dagger + Z_i + \text{h.c.} \right) . \tag{1}$$

The local clock operators $X_i$ and $Z_j$ commute on different sites, while on the same site they obey

$$Z^3 = X^3 = \mathbf{1}, \qquad Z^2 = Z^\dagger, \qquad X^2 = X^\dagger \quad \text{and} \quad ZX = \omega XZ, \quad \text{where} \quad \omega = e^{2\pi i/3}. \quad (2)$$

For future reference, we also define $Y = ZX$. Like a spin-1/2 chain can be written in terms of fermionic operators via a Jordan-Wigner transformation, 3-state Potts models can be written in terms of parafermion operators via a Fradkin-Kadanoff transformation [14, 24]. Introducing the lattice parafermion operators

$$\alpha_{2i-1} = \left(\prod_{j<i} Z_j\right) X_i, \qquad \alpha_{2i} = \omega \left(\prod_{j\leq i} Z_j\right) X_i, \quad (3)$$

that satisfy $\alpha_j^3 = 1$, $\alpha_j^\dagger = \alpha_j^2$ and $\alpha_i \alpha_j = \omega^{\text{sign}(i-j)} \alpha_j \alpha_i$, the Hamiltonian (1) takes the form

$$H_\pm = \pm \sum_{i=1}^{2L} \left(\omega \alpha_{2i+1}^\dagger \alpha_{2i} + \text{h.c.}\right) + \left(\omega \alpha_{2i}^\dagger \alpha_{2i-1} + \text{h.c.}\right). \quad (4)$$

While this model is quadratic in terms of the parafermion operators, it is strongly interacting and not solvable with a Fourier transformation. Due to the formal similarity to free, quadratic fermions, we refer to it describing 'quadratic' parafermions.[1] When higher order terms terms appear, such as four or six parafermion terms in the models we construct, we refer to the parafermions exhibiting many-body interactions.

The nature of the criticality of the 3-state Potts model depends on the overall sign. As shown in Table 1, $H_-$ is in the universality class of $Z_3$ parafermion CFT with central charge $c = 4/5$ [29, 30], while the criticality of $H_+$ is described by the $u(1)_6$ CFT with $c = 1$ [31–33]. The table also summarizes the new critical models we analytically construct and the CFTs that occur in them. Since they all involve terms such as $Z_i Z_{i+1}$, $Z_i Z_{i+1} Z_{i+2}$ or $Z_i Y_{i+1} Z_{i+2}$, that involve four or more parafermion operators, all the new critical models correspond to parafermions with many-body interactions.

To explain how these models were constructed, the paper is structured as follows. In Section 2.1 we introduce the key concept of modular invariant partition functions of CFTs that characterize the possible distinct critical behaviors. To illustrate these concepts and pave the way for constructing the generalized 3-state Potts models, in Section 2.2 we revisit the relation between two critical transverse field Ising chains and the XY chain from the perspective of modular invariant partition functions. This example, as well as its generalizations, has been studied in our previous works [17, 18] from the anyon condensation perspective, whose connection to the current approach of modular invariants is explained in Section 2.3. In Section 3 we review the $Z_3$ parafermion and $u(1)_6$ criticalities that appear in the 3-state Potts model and then in Section 4 we provide the detailed derivations of the new models listed in Table 1. Section 5 discusses the possible generalization of our construction to $k$-state Potts models and we conclude with Section 6. A detailed discussion on the fusion rule symmetries that are useful in constructing non-diagonal permutation invariants can be found in Appendix A.

## 2 The Method: Modular invariant partition functions

Our construction is based on the properties of the partition function of the CFT describing a critical point. The partition function of every two-dimensional CFT on the torus must be

---

[1] Non-interacting parafermions should not be confused here with free parafermions [19]. When one considers the non-Hermitian 3-state Potts model, i.e. drops the Hermitian conjugate + h.c. from (1), the spectrum (containing complex eigenvalues), is free in the sense that all states can be constructed from the algebra of the parafermion operators $\alpha_i$.

Table 1: Summary of the results. Each Hamiltonian given by $H = \sum_i (O_i + \text{h.c.})$ in terms of the clock operators (2) is critical and described by a modular invariant CFT that has central charge $c$ and the listed number of primary fields. In the CFT column $Z_3$, $u(1)_6$ and $su(2)_3 = c(Z_3 \times u_1(6))$ denote the diagonal invariants corresponding to these CFTs, while $\pi(G)$ and $c(G)$ denote the non-trivial permutation and condensation invariants of the CFT $G$, respectively, and $G^{\times n}$ denotes a product of $n$ CFTs of type $G$. Apart from the model with the $su(2)_3$ critical point that has a two-site unit cell, all the models are translationally invariant.

| $O_i$ | CFT | c | # primary fields |
|:---:|:---:|:---:|:---:|
| $-X_i^\dagger X_{i+1} - Z_i$ | $Z_3$ | 4/5 | 6 |
| $+X_i^\dagger X_{i+1} + Z_i$ | $u(1)_6$ | 1 | 6 |
| $-X_i^\dagger X_{i+1} - Z_i Z_{i+1}$ | $\pi(Z_3^{\times 2})$ | 8/5 | $6^2$ |
| $+X_i^\dagger X_{i+1} + Z_i Z_{i+1}$ | $\pi(u(1)_6^{\times 2})$ | 2 | $6^2$ |
| $-X_{2j-1}^\dagger X_{2j} - Z_{2j} Z_{2j+1} + 2X_{2j}^\dagger X_{2j+1} + 2Z_{2j-1} Z_{2j}$ | $su(2)_3$ | 9/5 | 4 |
| $-X_i^\dagger X_{i+1} - Z_i Z_{i+1} Z_{i+2}$ | $c(Z_3^{\times 3})$ | 12/5 | 24 |
| $+X_i^\dagger X_{i+1} + Z_i Z_{i+1} Z_{i+2}$ | $c(u(1)_6^{\times 3})$ | 3 | 24 |
| $-X_i^\dagger X_{i+1} - Z_i Y_{i+1} Z_{i+2}$ | $\pi(Z_3^{\times 3})$ | 12/5 | $6^3$ |
| $+X_i^\dagger X_{i+1} + Z_i Y_{i+1} Z_{i+2}$ | $\pi(u(1)_6^{\times 3})$ | 3 | $6^3$ |

invariant under symmetry operations known as modular transformations [2, 28]. In the current setting of one-dimensional chains, this means that the spectrum of a chain with periodic boundary conditions is described (in the thermodynamic limit) by a modular invariant partition function. It is possible that a given CFT admits several distinct modular invariant partition functions, in which case each describes critical behavior of distinct type (see [34–37] for some early references). Often in literature, due to these appearing most often in physically relevant systems, one implicitly refers to the diagonal partition function when referring to a particular critical point being described by a given CFT. However, in general this is too restrictive. There can also be other non-diagonal modular invariant partition functions, to which we refer to as permutation or condensation invariants for reasons to be explained below. We show that also these can in fact be realized in microscopic critical models. Before we do so, we should mention that many important properties, do not depend on the precise modular invariant, or even the boundary conditions. Perhaps the most important such property is the specific heat, which is determined by the central charge of the CFT. Nevertheless, many properties do depend on the details of (at least the low-lying part of) the spectrum, such as finite temperature and dynamical properties.

To show how different modular invariants can be realized, we start from a known critical microscopic model described by the diagonal invariant of a CFT and find all the non-diagonal modular invariant partition functions by employing additional symmetries of the CFT. Then we express these partition functions as linear combinations of the diagonal partition function when it is restricted to different symmetry sectors and have different boundary conditions. This linear combination is subsequently translated into a Hamiltonian term that, when added to the initial critical microscopic system described by the diagonal invariant, changes the nature of the criticality to the non-diagonal invariant. While the resulting system is in general non-local, in all cases we are able to find canonical duality transformations that give a local and translational invariant representative of the new critical model.

To demonstrate this method in detail, in this section we first review the modular invariance of CFT partition functions, and comment on the relation with extended algebras and the orbifold construction. Then, by revisiting our earlier work [17, 18], we illustrate the method

by showing how the critical XY model can be derived from two decoupled critical transverse field Ising chains. We also review the connection to anyon condensation that provides a simple criterium to predict when a given CFT admits particular type of non-diagonal invariant known as a condensation invariant.

## 2.1 Modular invariant partition functions

The partition function of a system described by the Hamiltonian $H$ is given by $Z = \text{Tr}(e^{-\beta H})$. When the system is critical, the Hamiltonian can be expanded in the Virasoro algebra generators $L_i$ satisfying the Virasoro algebra with central charge $c$ [2]. A special role is played by the operator $L_0$. Its eigenvalues take the form $\epsilon = h_{\phi_i} + n$, where $n$ a non-negative integer, that gives the energies of each state in the spectrum. All the energies are shifted away from integer values by $h_{\phi_i}$, the scaling dimensions of a each chiral primary field $\phi_i$. The distinct primary fields $\phi_i$, whose number is finite in all the cases we consider, are specific to the CFT describing the critical point. Akin to usual symmetry sectors, every state that descends from $\phi_i$ belongs to the same 'conformal tower' that can be viewed as a sector of the CFT. The partition functions of these sectors are the chiral characters associated with different primary fields $\phi_i$. Formally they are defined as the polynomials

$$\chi_{\phi_i} = q^{-c/24} \text{Tr}_{\phi_i} q^{L_0} = q^{-c/24} q^{h_{\phi_i}} \sum_{n=0,1,\dots} a_n q^n \,, \tag{5}$$

where the trace is over all the eigenstates of $L_0$ belonging to the conformal tower associated with the primary field $\phi_i$. Here $q = e^{2\pi i \tau}$ is a formal variable of the modular parameter $\tau$ whose powers are the energies $\epsilon$ of the states and the non-negative integers $a_n$ encode their degeneracy.

The primary field operators are in general chiral, which means that in 1+1D they can either propagate left or right. This means that a single chiral CFT can describe chiral states, such as edge states of 2+1D topologically ordered states, but not genuine 1D states that arise from local Hamiltonians (chiral operators do not have local representations). Instead, 1+1D critical systems are described by combining the two chiral halves of the CFT. The full partition function describing such systems can be written as $Z = (q\bar{q})^{-c/24} \text{Tr}\, q^{L_0} \bar{q}^{\bar{L}_0}$, where the bar denotes Hermitian conjugation. In terms of the left and right chiral characters $\chi_\phi$ and $\bar{\chi}_\phi$, the partition function of a 1+1D critical system takes the general form

$$Z = \sum_{\phi_1, \phi_2} M_{\phi_1, \phi_2} \chi_{\phi_1} \bar{\chi}_{\phi_2} \,, \tag{6}$$

where $\phi_1$ and $\phi_2$ are summed over all primary fields of the CFT and $M_{\phi_1, \phi_2}$ are non-negative integers.

Possible reparametrizations of the torus set stringent constraints on the allowed partition functions of a given CFT described by the matrices $M$ with elements $M_{\phi_1, \phi_2}$. These *modular transformations* formally transform the modular parameter $\tau$ as

$$S: \quad \tau \to -1/\tau, \qquad T: \tau \to \tau + 1. \tag{7}$$

Modular invariance of the partition function is then equivalent to demanding that

$$[M, S] = [M, T] = 0, \tag{8}$$

where $S$ and $T$ are $n \times n$ modular matrices specific to a given CFT with $n$ primary fields. Formally, they are obtained by studying the behavior of the chiral characters (5) under the modular transformations (7), but for most CFTs they can be found in the literature. The matrix

$T$ is always diagonal with entries $T_{\phi_1,\phi_2} = e^{2\pi i(h_{\phi_1}-c/24)}\delta_{\phi_1,\phi_2}$, while $S$ takes a form that does not have a simple expression in terms of only the scaling dimensions $h_{\phi_i}$. To find all matrices $M$ subject to the constraints (8) amounts to classifying all modular invariant partition functions and hence all distinct critical behaviors that can occur in 1+1D systems. This, however, is a challenging task. A complete classification has been achieved only in a limited set of cases, such as the minimal models [37], but a general classification for arbitrary CFTs is still lacking. For a CFT with a small number of primary fields, one can perform a numerical brute-force search, which is the approach we take here.

The different possible modular invariant partition functions fall into different classes. First, clearly the identity matrix $M_{\phi_1,\phi_2} = \delta_{\phi_1,\phi_2}$ commutes with both $S$ and $T$. This is known as the *diagonal modular invariant*, which is the most common partition function associated with a given CFT. In the literature, a 1+1D critical systems being described by a CFT usually refers to this invariant. The second case corresponds to the matrix $M$ being block-diagonal. Using the original characters $\chi_{\phi_i}$, one constructs new ones, $\chi'_j = \chi_{\phi_{j_1}} + \chi_{\phi_{j_2}} + \cdots$, which are combined 'in a diagonal way' to form a new partition function $Z' = \sum_j \chi'_j \bar{\chi}'_j$. Typically, not all the original characters appear in the block diagonal partition function, and it can happen that certain characters appear several times. For the purpose of this paper, we call these block-diagonal invariants *condensation modular invariants* due to their connection to anyon condensation explained in Section 2.3. One can view the new chiral characters $\chi'_j$ as corresponding to a primary field of a CFT with the same central charge as the original one, but with a different field content with generically less primary fields. Thus criticality described by a condensation invariant of some CFT is always equivalent to the diagonal invariant of some other CFT. We note that there is direct relation between condensation invariants and CFTs with an extended chiral algebra. If the new vacuum character $\chi'_0 = \chi_0 + \sum_j \chi_j$, the primaries related to the $\chi_j$ (which in the current paper, are chiral products of Virasoro characters) have integer scaling dimensions, that is, they are currents. Indeed, in the CFT corresponding to the condensation invariant, the chiral algebra has been extended with these currents. For more information about CFTs with extended algebras, we refer to [38].

Given the diagonal invariant, it is sometimes possible to construct a *permutation modular invariant*. It is of the form $M_{\phi_1,\phi_2} = \delta_{\phi_1,\pi(\phi_2)}$, where $\pi(\phi)$ denotes a permutation of the primary fields, which leaves the fusion rules of the primary fields invariant (we review fusion rule symmetries in Appendix A). In other words, all chiral characters of both chiral halves appear, but $M$ is no longer the identity matrix, but a permutation matrix (thus, in this case, the chiral algebra is not extended). Depending on the symmetries of the CFT, it may or may not give rise to the same partition function as the diagonal one when (6) is written out as a polynomial in $q$ and $\bar{q}$. If not, then while the primary field content in each chiral half is the same as for the diagonal invariant, the local physical observables and the energy spectrum are different due to them being constructed from the left and right moving components of different primary fields. While criticality corresponding to permutation invariants has been little studied in the context of physical models, in this work we show that they can indeed also arise in local systems. We note that to construct permutation invariants, one does not have to start from the diagonal invariant. One can also obtain different invariants by performing a permutation of the fields on a block-diagonal (or condensation) invariant.

## 2.2 Example: Modular invariants for products of Ising CFTs

To illustrate these concepts, we give an explicit example of a case where distinct modular invariants appear – a product theory of two Ising CFTs. This example also illustrates how to deal with product CFTs and how the partition functions depend on twisted boundary conditions, which are the key methods for constructing new critical Potts-like models later. We refer

to [39] for more information on the relation between the Ising$^{\times 2}$ and $u(1)_4$ CFTs that we will now discuss.

The Ising CFT has three primary fields, denoted by $\mathbf{1}$, $\sigma$ and $\psi$, with scaling dimensions $h_{\mathbf{1}} = 0$, $h_\sigma = 1/16$ and $h_\psi = 1/2$. The $S$ and $T$ matrices are represented by

$$S = \frac{1}{2}\begin{pmatrix} 1 & \sqrt{2} & 1 \\ \sqrt{2} & 0 & -\sqrt{2} \\ 1 & -\sqrt{2} & 1 \end{pmatrix}, \qquad T = e^{\frac{\pi i}{24}}\begin{pmatrix} 1 & 0 & 0 \\ 0 & e^{\frac{\pi i}{8}} & 0 \\ 0 & 0 & e^{\pi i} \end{pmatrix}. \tag{9}$$

For the Ising CFT, there is only a a single modular invariant partition function given by the diagonal invariant $M = \mathbf{1}_{3\times 3}$.

The product of two Ising CFTs, that we call Ising$^{\times 2}$ CFT, has $c = 1$ and nine primary fields that we will denote by pair of labels $(\mathbf{1}, \mathbf{1}); (\mathbf{1}, \sigma); (\mathbf{1}, \psi); \dots$ spanning all the possible combinations of the primary fields. The scaling dimensions for these product primary fields are additive in the constituent fields, i.e. $h_{(\phi_1,\phi_2)} = h_{\phi_1} + h_{\phi_1}$. Similarly, their chiral characters are simply products $\chi_{(\phi_1,\phi_2)} = \chi_{\phi_1}\chi_{\phi_2}$ and the representations of the modular matrices are given as the tensor products $S_{\text{Ising}^{\times 2}} = S \otimes S$ and $T_{\text{Ising}^{\times 2}} = T \otimes T$. The matrix $M$ is thus now a $9 \times 9$ matrix, but it is still straightforward to find all the modular invariant partition functions satisfying (8) by a brute force calculation. It turns out that there are three solutions given by

$$Z_{\text{Ising}^{\times 2}} = \sum_{\phi_1,\phi_2 \in \text{Ising}} |\chi_{(\phi_1,\phi_2)}|^2, \tag{10}$$

$$Z_{\text{Ising}^{\times 2}}^{\pi} = \sum_{\phi \in \text{Ising}} |\chi_{(\phi,\phi)}|^2 + \sum_{\phi_1 \neq \phi_2 \in \text{Ising}} \chi_{(\phi_1,\phi_2)}\bar{\chi}_{(\phi_2,\phi_1)}, \tag{11}$$

$$Z_{\text{Ising}^{\times 2}}^{u(1)_4} = |\chi_{(\mathbf{1},\mathbf{1})} + \chi_{(\psi,\psi)}|^2 + |\chi_{(\mathbf{1},\psi)} + \chi_{(\psi,\mathbf{1})}|^2 + 2|\chi_{(\sigma,\sigma)}|^2. \tag{12}$$

The $Z_{\text{Ising}^{\times 2}}$ is the diagonal invariant partition function of the Ising$^{\times 2}$ CFT, while $Z_{\text{Ising}^{\times 2}}^{\pi}$ is a permutation invariant. In the case of Ising$^{\times 2}$ it is not independent, but gives rise to a partition function that is identical to the diagonal invariant. This is due to the 'layer' symmetry of Ising$^{\times 2}$ that leaves the theory invariant under relabeling of the primary fields $(\phi_1, \phi_2) \to (\phi_2, \phi_1)$, i.e. $\chi_{(\phi_2,\phi_1)} = \chi_{(\phi_1,\phi_2)}$, so that $Z_{\text{Ising}^{\times 2}}^{\pi} = Z_{\text{Ising}^{\times 2}}$.

On the other hand, the block diagonality and the absence of some primary fields in $Z_{\text{Ising}^{\times 2}}^{u(1)_4}$ means that it is a condensation invariant. If one identifies

$$\chi_{\tilde{\mathbf{1}}} = \chi_{(\mathbf{1},\mathbf{1})} + \chi_{(\psi,\psi)}, \quad \chi_{\tilde{\psi}} = \chi_{(\mathbf{1},\psi)} + \chi_{(\psi,\mathbf{1})} \quad \text{and} \quad \chi_\lambda = \chi_{\bar{\lambda}} = \chi_{(\sigma,\sigma)}, \tag{13}$$

one finds that it should correspond to a diagonal invariant of a theory with three non-trivial primary fields with scaling dimensions $h_{\tilde{\psi}} = \frac{1}{2}$ and $h_\lambda = h_{\bar{\lambda}} = \frac{1}{8}$. This CFT, which is a $c = 1$ CFT of a compactified boson, is called $u(1)_4$. In fact, this was to be expected since it is well known that the Ising$^{\times 2}$ CFT is obtained from the $u(1)_4$ CFT by means of an orbifold construction [40]. Going in the other direction, one can obtain the $u(1)_4$ CFT form the Ising$^{\times 2}$ theory by 'adding the bosonic field $(\psi, \psi)$ to the chiral algebra'. We describe this relation between these CFTs from the perspective of anyon condensation below.

We have thus found that starting from the chiral Ising$^{\times 2}$ CFT, one can construct both the diagonal invariant and condensation invariant partition functions. Since it is well known that the critical transverse field Ising (TFI) chain is described by Ising CFT [28], the criticality corresponding to the diagonal invariant of Ising$^{\times 2}$ is clearly realized in a system of two decoupled TFI chains. The corresponding microscopic Hamiltonian is given, for instance, as the critical TFI chain with nest-nearest exchange interactions

$$H_{\text{Ising}^{\times 2}} = \sum_{i=1}^{L} \left(\sigma_i^x \sigma_{i+2}^x + \sigma_i^z\right), \tag{14}$$

where $\sigma_i^\alpha$ are the usual spin-1/2 Pauli matrices.

To write down a microscopic model for the condensation invariant, we begin by studying how the diagonal partition function of a single Ising CFT depends on the symmetry sectors and boundary conditions. The symmetry sectors are inherited from the TFI chain that has $Z_2$ spin flip symmetry and thus two symmetry sectors that we label by $Q = 0, 1$. A TFI chain can also have either periodic or anti-periodic boundary conditions, which we denote by $\tilde{Q} = 0, 1$, respectively. Denoting the partition function in symmetry sector $Q$ with boundary conditions $\tilde{Q}$ by $Z_Q^{\tilde{Q}}$, it is well known that [41]

$$
\begin{aligned}
Z_0^0 &= \chi_\mathbf{1} \bar{\chi}_\mathbf{1} + \chi_\psi \bar{\chi}_\psi & Z_1^0 &= \chi_\sigma \bar{\chi}_\sigma, \\
Z_0^1 &= \chi_\sigma \bar{\chi}_\sigma & Z_1^1 &= \chi_\mathbf{1} \bar{\chi}_\psi + \chi_\psi \bar{\chi}_\mathbf{1},
\end{aligned}
\tag{15}
$$

where there holds $Z_Q^{\tilde{Q}} = Z_{\tilde{Q}}^Q$ as required by the duality between the symmetry sectors and boundary conditions when solving the transverse field Ising model with a Jordan-Wigner transformation [42]. Clearly, summing over the symmetry sectors with periodic boundary conditions gives the diagonal invariant partition function of the Ising CFT, $Z_\text{Ising} = Z_0^0 + Z_1^0 = \chi_\mathbf{1} \bar{\chi}_\mathbf{1} + \chi_\psi \bar{\chi}_\psi + \chi_\sigma \bar{\chi}_\sigma$.

Likewise, the Hamiltonian (14) for two decoupled TFI chains has $Z_2 \times Z_2$ symmetry and the boundary conditions can be independently chosen for both chains. Employing the property $\chi_{(\phi_1, \phi_2)} = \chi_{\phi_1} \chi_{\phi_2}$ for product CFTs, one can then verify that the diagonal invariant of the $\text{Ising}^{\times 2}$ CFT is obtained by summing over the four symmetry sectors with periodic boundary conditions

$$
Z_{\text{Ising}^{\times 2}} = Z_0^0 Z_0^0 + Z_0^0 Z_1^0 + Z_1^0 Z_0^0 + Z_1^0 Z_1^0 .
\tag{16}
$$

Instead of decoupled TFI chains, consider a coupled system where the boundary condition of one the Ising CFTs is given by the symmetry sector of the other, and vice versa. Employing the explicit forms of the partition functions (15), one can verify that one obtains now precisely the condensation invariant partition function (12) when summing over all symmetry sectors

$$
Z_{\text{Ising}^{\times 2}}^{u(1)_4} = Z_0^0 Z_0^0 + Z_0^1 Z_1^0 + Z_1^0 Z_0^1 + Z_1^1 Z_1^1.
\tag{17}
$$

To implement this coupling in the decoupled Hamiltonian (14), we introduce the coupling boundary Hamiltonian

$$
H_B = (\mathcal{P}_e - \mathbf{1}) \sigma_{L-1}^x \sigma_1^x + (\mathcal{P}_o - \mathbf{1}) \sigma_L^x \sigma_2^x,
\tag{18}
$$

where $\mathcal{P}_e = \prod_{i=1}^{L/2} \sigma_{2i}^z$ and $\mathcal{P}_o = \prod_{i=1}^{L/2} \sigma_{2i-1}^z$ are the independent symmetry operators on even and odd sites, respectively. The Hamiltonian $H_{\text{Ising}^{\times 2}} + H_B$ correlates then the boundary conditions and symmetry sectors of the two chains precisely as done in the linear combination (17) for the condensation invariant partition function. By construction the resulting system must thus be critical and described by the $u(1)_4$ CFT. While this Hamiltonian is manifestly non-local, we have shown in earlier work [17, 18] that by using non-local duality transformations it can be mapped precisely to the critical XY chain $H_{XY} = \sum_i \sigma_i^x \sigma_{i+1}^x + \sigma_i^y \sigma_{i+1}^y$ that is known to be described by the diagonal invariant of the $u(1)_4$ CFT. We mention that the term $H_B$ effectively causes the boundary conditions to be 'twisted'. In contrast to the usual, well-studied, twisted boundary conditions, in our case, the 'twist' in the boundary conditions is symmetry sector dependent.

The example we just described, allows us to comment on the relation between the construction and orbifolding. It is well known that the $\text{Ising}^{\times 2}$ CFT is the $Z_2$ orbifold of the $u(1)_4$ CFT. So, by constructing the condensation modular invariant, one is effectively doing an orbifold

construction in reverse. On the level of microscopic hamiltonians, the 'condensation direction' is the more straightforward one, because one starts by a simple doubling. Realizing that $H_{XY}$ can be written as $H_{\text{Ising}^{\times 2}} + H_B$ after a non-local canonical transformation is in general much harder.

We have demonstrated how finding out all possible modular invariant partition functions enables to determine whether a given critical system enables to construct new microscopic models for the non-diagonal partition functions. Before applying this strategy to construct microscopic models for criticality in Potts-type models, we review briefly the connection to anyon condensation that provides an easy way to predict when such constructions might be possible without knowing all the modular invariant partition functions (which in general is a hard task for CFTs with increasing number of primary fields).

## 2.3 Anyon condensation perspective

There is an intriguing connection between the modular invariant partition functions and anyon condensation [43,44]. Chiral CFTs describe also the 1D edge states of 2D topologically ordered states of matter, with the possible anyonic quasi-particle excitations being in one-to-one correspondence with the primary fields of the CFT. Via this bulk-edge correspondence the anyonic statistics of the quasi-particles are given by the scaling dimensions that are interpreted as their topological spins. If for some quasi-particle $a$ the spin $h_a$ is an integer, then it may condense, which results in general in a transition to another topologically ordered state of matter. As this state has a different quasi-particle content in the bulk, it must also have a different CFT describing the edge and hence the framework of anyon condensation can in general be expected to also predict relations between CFTs themselves without referring to a particular realization of 2D topological order.

Indeed, in our previous works [17,18] we have shown that there is a precise counterpart. This insight enabled us to derive microscopic spin chains for all criticalities in the $so(N)_1$ universality class. The example considered in Section 2.2 is the simplest case of this hierarchy ($u(1)_4 \simeq so(2)_1$) that also has a realization in generalized cluster models [6]. In particular, we showed that by correlating boundary conditions with symmetry sectors, those primary fields that are predicted to be confined in a condensation transition can be removed from the spectrum and that this changes the critical description precisely as predicted by the framework of anyon condensation [43,44]. In our example above, we employed this same correlating of symmetry sectors and boundary conditions to construct the modular invariant partition function (12). In the condensation language the block diagonality of $Z^{u(1)_4}_{\text{Ising}^{\times 2}}$ resulting in the diagonal partition function of $u(1)_4$ CFT via the identifications (13) can be understood as follows. The boson $(\psi, \psi)$ with $h_{(\psi,\psi)} = 1$ condensed and thus became indistinguishable from the vacuum $(\mathbf{1}, \mathbf{1})$. This results in the confinement of the primary fields $(\sigma, \mathbf{1})$, $(\sigma, \psi)$, $(\mathbf{1}, \sigma)$ and $(\psi, \sigma)$ that do not appear in the block-diagonal partition function. The consistency of the resulting theory requires that $(\psi, \mathbf{1})$ and $(\mathbf{1}, \psi)$ are treated as the same new fermionic field $\tilde{\psi}$ and that the field $(\sigma, \sigma)$ must split into two distinct fields $\lambda$ and $\bar{\lambda}$.

The advantage of the condensation picture is that even if not all modular invariant partition functions are known, the existence of a primary field with an integer scaling dimension that is also a simple current implies that a condensation is possible and hence a block-diagonal modular invariant partition function exists[2]. As it is easy to figure out which fields should be confined [43,44], this gives a principle how to construct condensation invariant partition functions by correlating symmetry sectors and boundary conditions and subsequently verify

---

[2]If primary field with an integer scaling dimension is not a simple current, i.e. it does not have Abelian fusion rules with itself and with all other primary fields, the situation is more complicated the existence of block-diagonal invariants needs to be studied on a case-by-case basis [43–46]

that modular invariance is indeed satisfied. We turn now to apply these ideas to construct generalized 3-state Potts models whose criticalities are described by non-diagonal modular invariants of either the permutation or the condensation type.

## 3 Building block: Critical behavior of the 3-state Potts model

In the example of Section 2.2, we started from decoupled TFI chains to construct the XY model corresponding to the $u(1)_4$ condensation invariant. To construct generalized critical 3-state Potts models, we follow the same strategy, but start from some number of decoupled 3-state Potts models. In this section we review the $Z_3$ parafermion and $u(1)_6$ criticality that appear in the 3-state Potts model (1) for different overall signs. We also summarize how the partition function depends on symmetry sectors and boundary conditions, which is needed for our construction.

For negative overall sign, the 3-state Potts model is described by the diagonal invariant of the $Z_3$ parafermion CFT [30], while for positive sign it is described by the diagonal invariant of the $u(1)_6$ CFT [33]. For both cases these are the only independent modular invariants. To describe the field content of these CFTs on equal footing, it is convenient to view the $Z_k$ parafermion CFT as the coset $su(2)_k/u(1)_{2k}$, which holds for any positive integer $k$ [29]. This enables to describe the primary field content of $Z_k$ theories in terms of the simpler field content of $su(2)_k$ and $u(1)_{2k}$ CFTs. For generality, we provide these properties for generic $k$, although in this work we mainly focus on the case $k = 3$.

The $su(2)_k$ CFT has central charge $c = \frac{3k}{k+2}$ and contains $k + 1$ primary fields [2]. We denote these by $\varphi^l$ for $l = 0, 1, \ldots, k$. The scaling dimension of each primary field is given by $h^l = \frac{l(l+2)}{4(k+2)}$ and they obey the fusion rules

$$\varphi^{l_1} \times \varphi^{l_2} = \varphi^{|l_1-l_2|} + \varphi^{|l_1-l_2|+1} + \cdots + \varphi^{\max(l_1+l_2, 2k-l_1-l_2)} . \tag{19}$$

For example, one of the new models we construct has a $su(2)_3$ critical point, as shown in Table 1. This $c = 9/5$ CFT has four primary fields with scaling dimensions $\{0, 3/20, 2/5, 3/4\}$.

On the other hand, the $u(1)_{2k}$ CFT is the $c = 1$ compactified chiral boson theory with $2k$ primary fields [2]. The primary fields are labeled $\varphi_m$, where $m$ is an integer defined modulo $2k$. Here we choose the convention that $m$ lies in the range $-k < m \leq k$, which means that the scaling dimensions are given by $h_m = \frac{m^2}{2k}$ and the fusion rules can be written as

$$\varphi_{m_1} \times \varphi_{m_2} = \varphi_{m_1+m_2} . \tag{20}$$

For example, for the $u(1)_6$ CFT describing the 3-state Potts model for a positive overall sign, there are six primary fields with scaling dimensions $\{0, 1/6, 1/6, 2/3, 2/3, 3/2\}$.

The $Z_k$ parafermion CFT has the central charge $c = \frac{2(k-1)}{k+2}$ and it is equivalent to the coset $su(2)_k/u(1)_{2k}$. Its primary fields are then labeled by the labels of the $su(2)_k$ and $u(1)_{2k}$ theories, namely $\varphi_m^l$. In addition to the constraints on $l$ and $m$ we already introduced, due to the coset the labels also have to satisfy the additional constraint $l + m = 0 \bmod 2$ and the identification $\varphi_m^l \equiv \varphi_{m+k}^{k-l}$. Using this, one can choose labels $\phi_m^l$ such that $|m| \leq l$, in which case the scaling dimensions of the $k(k+1)/2$ primary fields are given by $h_m^l = h^l - h_m$. The fusion rules follow directly from the fusion rules of the $su(2)_k$ and $u(1)_{2k}$ theories and are given by

$$\varphi_{m_1}^{l_1} \times \varphi_{m_2}^{l_2} = \varphi_{m_1+m_2}^{|l_1-l_2|} + \varphi_{m_1+m_2}^{|l_1-l_2|+1} + \cdots + \varphi_{m_1+m_2}^{\max(l_1+l_2, 2k-l_1-l_2)} . \tag{21}$$

For the $Z_3$ parafermion CFT describing the 3-state Potts model for a negative overall sign, there are six primary fields with scaling dimensions $\{0, 2/3, 2/3, 2/5, 1/15, 1/15\}$. In the literature

these are often also labeled as $\{\psi_0 = \varphi_0^0, \psi_1 = \varphi_2^0, \psi_2 = \varphi_{-2}^0, \tau_0 = \varphi_0^2, \tau_1 = \varphi_2^2, \tau_2 = \varphi_{-2}^2\}$. For completeness, we note that the modular $T$ matrix is determined by the scaling dimensions as discussed in Section 2.1, and the $S$ matrix is given by

$$
S = \frac{1}{\sqrt{3+6\phi}}
\begin{pmatrix}
1 & 1 & 1 & \phi & \phi & \phi \\
1 & \omega & \omega^2 & \phi & \omega\phi & \omega^2\phi \\
1 & \omega^2 & \omega & \phi & \omega^2\phi & \omega\phi \\
\phi & \phi & \phi & -1 & -1 & -1 \\
\phi & \omega\phi & \omega^2\phi & -1 & -\omega & -\omega^2 \\
\phi & \omega^2\phi & \omega\phi & -1 & -\omega^2 & -\omega
\end{pmatrix},
\tag{22}
$$

where $\omega = e^{2\pi i/3}$ and $\phi = (1+\sqrt{5})/2$. The ordering of the fields is $(\varphi_0^0, \varphi_2^0, \varphi_{-2}^0, \varphi_0^2, \varphi_2^2, \varphi_{-2}^2)$.

While the fusion rules are not of paramount importance to our construction, we have have included them to highlight a possible ambiguity in using the labels $l$ and $m$ to label the primary fields $\varphi_m^l$. If a permutation of the labels leaves the fusion rules invariant, i.e. there is a *fusion rule symmetry*, then the partition function corresponding to the permuted labeling may or may not be identical to the diagonal one. This can be checked by writing out the partition functions (6) as polynomials in $q$ and $\bar{q}$ and see whether two different labelings give rise to the same partition function or not. Thus the fusion rule symmetries provide easy ansatzes to try and see whether a given CFT admits non-diagonal partition functions. For instance, for the $Z_3$ parafermion CFT with primary fields $\varphi_m^l$, where $l = 0, 2$ and $m = -2, 0, 2$, the only permutation of the fields that leaves the fusion rules invariant is $\varphi_m^l \to \varphi_{-m}^l$. However, because the characters associated with $\varphi_m^l$ and $\varphi_{-m}^l$ are identical, this permutation gives again the diagonal partition function. The fusion rule symmetries for $Z_3$ and $u(1)_6$ CFTs, and for their product theories relevant to the present work, are discussed in more detail in Appendix A.

## 3.1 Partition functions for different symmetry sectors and boundary conditions

The dependence of the partition functions on the symmetry sectors and twisted boundary conditions is known for the $Z_k$ parafermion and the $u(1)_{2k}$ CFTs [41, 47]. A $k$-state Potts model always has a $Z_k$ symmetry described by the operators $\mathcal{P} = \prod_{i=1}^{L} Z_i$, where $Z^k = 1$ and the phase factor $\omega$ appearing in the commutation relations of the 3-state clock variables is replaced by $\omega = e^{\frac{2\pi i}{k}}$. This means that there are always $k$ symmetry sectors, that we label by $Q = 0, 1, \ldots, k-1$, corresponding to the eigenvalues $\omega^Q$ of $\mathcal{P}$. Similarly, there are $k$ possible twisted boundary conditions $\psi(x+L) = \omega^{\tilde{Q}}\psi(x)$ that we label by $\tilde{Q} = 0, 1, \ldots, k-1$.

Using this notation, the partition function for symmetry sector $Q$ with twisted boundary conditions $\tilde{Q}$ is given compactly by [47]

$$
Z_Q^{\tilde{Q}} = \frac{1}{2} \sum_{\substack{0 \le l \le k \\ -k < m \le k \\ l+m \bmod 2 = 0 \\ m-\tilde{Q} \bmod k = Q}} \chi_{\varphi_m^l} \bar{\chi}_{\varphi_{m-2\tilde{Q}}^l},
\tag{23}
$$

where the factor $\frac{1}{2}$ accounts for the double counting due to the identification $\varphi_{m+k}^{k-l} \equiv \varphi_m^l$. We note that the effect of twisting the boundary conditions only involves changing the labels $m$ of the parafermionic fields $\varphi_m^l$. In the case of the $u(1)_{2k}$ CFT, the partition functions for fixed symmetry sector and boundary condition take a completely analogous form. One simply drops the label $l$ from the field to obtain $Z_Q^{\tilde{Q}} = \sum_{-k < m \le k} \chi_{\varphi_m} \bar{\chi}_{\varphi_{m-2\tilde{Q}}}$, where the sum is constraint by $m - \tilde{Q} \bmod k = Q$.

Analogous to the example presented in Section (2.2), to construct critical generalizations of the Potts models we start from several copies of either the 3-state Potts models at either $Z_3$

or $u(1)_6$ critical point, or mixtures of them. The criticality of such systems is then described by products of the respective CFTs. The primary field content of product theories is obtained simply by taking all possible products of the primary fields of each CFT in the product. The scaling dimensions of these products are given as the sum of the scaling dimensions of the constituent fields and the fusion rules are just direct products of their fusion rules. Since the symmetries sectors $Q_n$ and boundary conditions $\tilde{Q}_n$ of each CFT in the product are independent, the partition function of a product theory of $N$ CFTs in a given symmetry sector and for given boundary conditions is given simply as the product $Z_{Q_1,\dots,Q_N}^{\tilde{Q}_1,\dots,\tilde{Q}_N} = Z_{Q_1}^{\tilde{Q}_1} \cdots Z_{Q_N}^{\tilde{Q}_N}$. For instance, a product of two $Z_3$ parafermion CFTs has 36 primary fields $(a_1, a_2)$ with scaling dimensions $h_{(a_1,a_2)} = h_{a_1} + h_{a_2}$. There are nine symmetry sectors labeled by $Q_1, Q_2 = 0, 1, 2$ with nine independent boundary conditions $\tilde{Q}_1, \tilde{Q}_2 = 0, 1, 2$. For each choice of them the partition function is given by $Z_{Q_1}^{\tilde{Q}_1} Z_{Q_2}^{\tilde{Q}_2}$.

## 4 Generalized 3-Potts models for non-diagonal modular invariants

This section contains the detailed derivation of the critical generalizations of 3-state Potts models listed in Table 1. We consider products of up to three $Z_3$ parafermion and $u(1)_6$ CFTs that can be realized by the 3-state Potts model for different overall signs. To obtain the non-diagonal modular invariant partition functions, for products of two CFTs we determine all the distinct invariants by numerically solving the equations (8). For products of three CFTs, a brute force solution becomes challenging due to the size of the modular matrices. Instead, guided by the anyon condensation perspective (see Section 2.3) and the fusion rule symmetries (see Appendix A), we consider all possible boundary terms and determine which of these give rise to non-diagonal modular invariants. For each case, we write the invariant as a linear combination over symmetry sectors and boundary conditions (see Section 3.1) and, similar to the example discussed in Section 2.3, construct a Hamiltonian term that implements the change in the nature of criticality in the microscopic setting. Finally, we introduce canonical duality transformations to find unitarily equivalent local models for each non-diagonal modular invariant.

### 4.1 The permutation invariants of $Z_3^{\times 2}$ and $u(1)_6^{\times 2}$ CFTs

We begin by considering the doubled $Z_3^{\times 2}$ parafermion CFT that is realized by the next-nearest neighbour Hamiltonian

$$H_{Z_3^{\times 2}} = -\sum_{i=1}^{L} (X_i X_{i+2}^{\dagger} + Z_i + \text{h.c.}) , \tag{24}$$

that describes two critical decoupled 3-state Potts chains. By brute force calculation we find that there are altogether 16 different modular invariants $M$. As expected from $Z_3^{\times 2}$ CFT containing no primary fields with integer scaling dimensions (so there are no bosons to condense from the anyon condensation perspective), all the invariants are permutation invariants. However, because of the fusion rule symmetries, there are only two distinct modular invariant partition functions (see Appendix A for details). All invariants related by the 8 permutations

$$\pi(\varphi_{m_1}^{l_1}, \varphi_{m_2}^{l_2}) = (\varphi_{s_1 m_1}^{l_1}, \varphi_{s_2 m_2}^{l_2}) \qquad \pi(\varphi_{m_1}^{l_1}, \varphi_{m_2}^{l_2}) = (\varphi_{s_1 m_2}^{l_2}, \varphi_{s_2 m_1}^{l_1}) , \tag{25}$$

where $s_1 = \pm 1$ and $s_2 = \pm 1$, give rise to the same a partition function as the diagonal invariant of the $Z_3^{\times 2}$ CFT. On the other hand, the 8 permutations

$$\pi(\varphi_{m_1}^{l_1}, \varphi_{m_2}^{l_2}) = (\varphi_{s_1 m_2}^{l_1}, \varphi_{s_2 m_1}^{l_2}) \qquad \pi(\varphi_{m_1}^{l_1}, \varphi_{m_2}^{l_2}) = (\varphi_{s_1 m_1}^{l_2}, \varphi_{s_2 m_2}^{l_1}), \qquad (26)$$

give rise to a permutation invariant corresponding to the partition function

$$Z_{Z_3^{\times 2}}^{\pi} = \sum_{\substack{l_1,l_2=0,\ldots,3 \\ -3 < m_1,m_2 \leq 3}} \chi_{(\varphi_{m_1}^{l_1}, \phi_{m_2}^{l_2})} \bar{\chi}_{(\phi_{m_2}^{l_1}, \phi_{-m_1}^{l_2})}, \qquad (27)$$

where again $l + m = 0 \bmod 2$.

Employing the knowledge how the partition functions depend on the symmetry sectors and boundary conditions (23), it is straightforward to verify that this permutation invariant can be expressed as the linear combination

$$Z_{Z_3^{\times 2}}^{\pi} = \sum_{Q_1,Q_2=0,1,2} Z_{Q_1}^{-Q_2} Z_{Q_2}^{Q_1}. \qquad (28)$$

In other words, the boundary condition of the first chain is given by the symmetry sector of the second chain, and vice versa, but in a twisted manner. The Hamiltonian term implementing this coupling between the two 3-state Potts chains is given by

$$H_B = -\left(\mathcal{P}_2^{\dagger} - \mathbf{1}\right)X_{L-1}X_1^{\dagger} - \left(\mathcal{P}_1 - \mathbf{1}\right)X_L X_2^{\dagger} + \text{h.c.}, \qquad (29)$$

where $\mathcal{P}_n = \prod_{j=0}^{L/2-1} Z_{2j+n}$ are the independent symmetry operators for each chain. While the Hamiltonian $H_{Z_3^{\times 2}} + H_B$ is manifestly non-local and breaks translation invariance, it can be brought into a translationally invariant form with the duality transformations

$$Z_{2j-1} = \tilde{X}_{2j-1}^{\dagger} \tilde{X}_{2j}, \qquad\qquad Z_{2j} = \tilde{Z}_{2j-1} \tilde{Z}_{2j},$$
$$X_{2j-1} = \Big(\prod_{k<j} \tilde{Z}_{2k-1} \tilde{Z}_{2k}\Big)\tilde{Z}_{2j-1}, \qquad X_{2j} = \tilde{X}_{2j}\Big(\prod_{k>j} \tilde{X}_{2k-1}\tilde{X}_{2k}^{\dagger}\Big)\mathcal{P}_1, \qquad (30)$$

where $j = 1, 2, \ldots, L/2$. It is straightforward to verify that the dual operators $\tilde{X}_i$ and $\tilde{Z}_i$ satisfy again the clock algebra (2). In terms of them one obtains the Hamiltonian

$$H_{Z_3^{\times 2}}^{\pi} = H_{Z_3^{\times 2}} + H_B = -\sum_i \left(\tilde{X}_i \tilde{X}_{i+1}^{\dagger} + \tilde{Z}_i \tilde{Z}_{i+1} + \text{h.c.}\right), \qquad (31)$$

which is critical by construction and described by the non-trivial permutation invariant (27) of the doubled $Z_3$ parafermion CFT.

There is a clear difference in the energy spectrum of the criticality described by the diagonal or the permutation invariants $Z_3^{\times 2}$. Recalling that the lowest energy states of each conformal tower labelled by the scaling dimensions $(h_l, h_r)$ have energy $h_l + h_r$, Table 2 shows the degeneracies of the lowest lying states. For the diagonal invariant there are four conformal towers with scaling dimensions $(h_l, h_r) = (\frac{1}{15}, \frac{1}{15})$ and four with $(h_l, h_r) = (\frac{16}{15}, \frac{16}{15})$. These are absent for the permutation invariant and replaced by eight towers with $(h_l, h_r) = (\frac{1}{15}, \frac{16}{15})$ and $(h_l, h_r) = (\frac{16}{15}, \frac{1}{15})$, that correspond to non-diagonal combination of the chiral halves. This means that if one compares the spectra of these models, the lowest lying excitations that are present in $H_{Z_3^{\times 2}}$ are absent in $H_{Z_3^{\times 2}}^{\pi}$. We confirmed this behaviour by explicitly diagonalizing the Hamiltonian (31) for system sizes up to $L = 12$ sites, showing that the rescaled finite-size spectrum indeed clearly differs from the diagonal invariant and precisely corresponds to the non-trivial permutation invariant (27).

Table 2: Comparison of the CFT spectra between the diagonal and permutation invariants of the $Z_3^{×2}$ (Left) and $u(1)_6^{×2}$ (Right) CFTs. The numbers in the table show the degeneracy of the lowest lying states with energy $h_l + h_r$ corresponding to the conformal tower $(h_l, h_r)$.

| $(h_l,h_r)$ | $Z_3^{×2}$ | $\pi(Z_3^{×2})$ | $(h_l,h_r)$ | $u(1)_6^{×2}$ | $\pi(u(1)_6^{×2})$ |
|---|---|---|---|---|---|
| $(\frac{4}{3},\frac{4}{3})$ | 4 | 4 | $(\frac{3}{2},\frac{3}{2})$ | 1 | 1 |
| $(\frac{16}{15},\frac{16}{15})$ | 4 | - | $(\frac{13}{12},\frac{13}{12})$ | 4 | - |
| $(\frac{4}{5},\frac{4}{5})$ | 1 | 1 | $(\frac{5}{6},\frac{5}{6})$ | 4 | 4 |
| $(\frac{11}{15},\frac{11}{15})$ | 8 | 8 | $(\frac{3}{4},\frac{3}{4})$ | 2 | 2 |
| $(\frac{2}{3},\frac{2}{3})$ | 4 | 4 | $(\frac{2}{3},\frac{2}{3})$ | 4 | 4 |
| $(\frac{1}{15},\frac{16}{15});(\frac{16}{15},\frac{1}{15})$ | - | 8 | $(\frac{1}{12},\frac{13}{12});(\frac{13}{12},\frac{1}{12})$ | - | 8 |
| $(\frac{7}{15},\frac{7}{15})$ | 4 | 4 | $(\frac{5}{12},\frac{5}{12})$ | 8 | 8 |
| $(\frac{2}{5},\frac{2}{5})$ | 2 | 2 | $(\frac{1}{3},\frac{1}{3})$ | 4 | 4 |
| $(\frac{2}{15},\frac{2}{15})$ | 4 | 4 | $(\frac{1}{6},\frac{1}{6})$ | 4 | 4 |
| $(\frac{1}{15},\frac{1}{15})$ | 4 | - | $(\frac{1}{12},\frac{1}{12})$ | 4 | - |
| $(0,0)$ | 1 | 1 | $(0,0)$ | 1 | 1 |

## The permutation invariant of $u(1)_6^{×2}$ CFT

The $u(1)_6^{×2}$ CFT is realized by two decoupled critical Potts chains described by the Hamiltonian $-H_{Z_3^{×2}}$, i.e. just by changing the overall sign of the Hamiltonian (24). Again, there are no fields with integer scaling dimensions, so there are no condensation invariants, but after accounting for the fusion rule symmetries we find again a single independent permutation invariant partition function $Z^\pi_{u(1)_6^{×2}}$. While writing it explicitly out is not particularly illustrative, its form can be seen from Table 2, which shows how the energy spectrum of a critical system corresponding to it differs from one described by the diagonal invariant. As above, some of the lowest lying states of the diagonal invariant are missing for the permutation invariant, which exhibits also conformal towers corresponding to non-diagonal combinations of the chiral halves.

To construct a microscopic model that realizes the permutation invariant, we find that the same coupling (28) between the two chains is required. Since changing the overall sign in (24) does not affect the duality transformations, the construction in the previous subsection applies directly also here. Thus the permutation invariant partition function $Z^\pi_{u(1)_6^{×2}}$ is realized by the Hamiltonian $H^\pi_{u(1)_6^{×2}} = -H^\pi_{Z_3^{×2}}$.

## 4.2 The $su(2)_3$ condensation invariant of the $Z_3 \times u(1)_6$ CFT

The remaining case that involves only a product of two CFTs is the $Z_3 \times u(1)_6$ criticality that is realized by two decoupled 3-state Potts models with opposite signs. This case is slightly more tricky than the previous two cases. When comparing the spectrum of a microscopic Hamiltonian to the predictions of a conformal field theory, one has to take a non-universal velocity into account, which is accomplished by an overall rescaling of the spectrum. If one couples two identical microscopic chains, the velocities are identical and one does not have to worry about them. However, in general the velocities of the two chains can be different. The Bethe ansatz solution of the 3-state Potts model indicates that the velocity of the $Z_3$ model is twice the velocity of the $u(1)_6$ model [32].

We can correct for this difference in velocity, without changing the universal description by $Z_3 \times u(1)_6$ CFT, by hand and include a relative factor of two in the Hamiltonian, so that

both criticalities have the same velocity. Our starting point thus is the Hamiltonian

$$H_{Z_3 \times u(1)_6} = -\sum_{j=1}^{L/2} \left(Z_{2j-1} + X_{2j-1} X_{2j+1}^\dagger + \text{h.c.}\right) + 2\sum_{j=1}^{L/2} \left(Z_{2j} + X_{2j} X_{2j+2}^\dagger + \text{h.c.}\right). \qquad (32)$$

Unlike the two preceding cases, this CFT contains a primary field with an integer scaling dimension suggesting that a condensation invariant exists. Indeed, by numerically evaluating the modular invariants, we find only the condensation invariant

$$Z_{Z_3 \times u(1)_6}^{su(2)_3} = |\chi_{\varphi_0^0} \chi_{\varphi_0} + \chi_{\varphi_2^0} \chi_{\varphi_{-2}} + \chi_{\varphi_{-2}^0} \chi_{\varphi_2}|^2 + |\chi_{\varphi_0^2} \chi_{\varphi_3} + \chi_{\varphi_2^2} \chi_{\varphi_1} + \chi_{\varphi_{-2}^2} \chi_{\varphi_{-1}}|^2 \qquad (33)$$
$$|\chi_{\varphi_0^2} \chi_{\varphi_0} + \chi_{\varphi_2^2} \chi_{\varphi_{-2}} + \chi_{\varphi_{-2}^2} \chi_{\varphi_2}|^2 + |\chi_{\varphi_0^0} \chi_{\varphi_3} + \chi_{\varphi_2^0} \chi_{\varphi_1} + \chi_{\varphi_{-2}^2} \chi_{\varphi_{-1}}|^2.$$

Identifying the blocks as new primary fields with scaling dimensions derived from the constituent fields, we find that this condensation invariant corresponds to the diagonal invariant of the $su(2)_3$ CFT described in Section 3. This was to be expected, since the $Z_3$ parafermion CFT is equivalent to the coset $su(2)_3/u(1)_6$. Multiplying in an additional $u(1)_6$ factor essentially reverses the coset construction.

To obtain a local microscopic Hamiltonian whose critical point is described by the $su(2)_3$ CFT, we find again that one has to use the same coupling (28) and hence the same boundary term (29) as in the case of the two coupled $Z_3$ parafermion CFTs. Thus the same canonical transformations can be employed and we find the local Hamiltonian

$$H_{su(2)_3} = -\sum_{j=1}^{L/2} \left(\tilde{X}_{2j-1} \tilde{X}_{2j}^\dagger + \tilde{Z}_{2j} \tilde{Z}_{2j+1} + \text{h.c.}\right) + 2\sum_{j=1}^{L/2} \left(\tilde{X}_{2j} \tilde{X}_{2j+1}^\dagger + \tilde{Z}_{2j-1} \tilde{Z}_{2j} + \text{h.c.}\right), \qquad (34)$$

which is translationally invariant with respect to a unit cell of two sites.

## 4.3 The condensation invariants of $Z_3^{\times 3}$ and $u(1)_6^{\times 3}$ CFTs

Having constructed microscopic representatives for all different modular invariant partition functions that can be constructed from two critical 3-state Potts models, we now turn to consider non-diagonal modular invariants of three decoupled 3-state Potts models described by either tripled $Z_3^{\times 3}$ parafermion CFT or $u(1)_6^{\times 3}$ CFTs. Any mixed product of three CFTs yields only invariants that are products of those obtained above for doubled theories with an additional $Z_3$ or $u(1)_6$ theory. For instance, the only non-diagonal invariants of $Z_3^{\times 2} \times u(1)_6$ correspond to the condensation invariant $su(2)_3 \times Z_3$ or the permutation invariant $\pi(Z_3^{\times 2}) \times u(1)_6$.

We begin with the $Z_3^{\times 3}$ CFT, which is realized by the Hamiltonian

$$H_{Z_3^{\times 3}} = -\sum_{i=1}^{L} (X_i X_{i+3}^\dagger + Z_i + \text{h.c.}). \qquad (35)$$

The presence of third nearest neighbour interactions only means this Hamiltonian corresponds to three decoupled critical 3-state Potts chains. The $Z_3^{\times 3}$ CFT contains primary fields with scaling dimension $h = 2$ and hence one expects to find a condensation invariant. Indeed, we find, in addition to a permutation invariant to be considered in the next section, a condensation invariant that gives rise to the partition function

$$Z_{Z_3^{\times 3}}^c = \sum_{l_1=0,2} \sum_{l_2=0,2} \sum_{l_3=0,2} |\chi_{\varphi_0^{l_1}} \chi_{\varphi_0^{l_2}} \chi_{\varphi_0^{l_3}} + \chi_{\varphi_2^{l_1}} \chi_{\varphi_2^{l_2}} \chi_{\varphi_2^{l_3}} + \chi_{\varphi_{-2}^{l_1}} \chi_{\varphi_{-2}^{l_2}} \chi_{\varphi_{-2}^{l_3}}|^2 + \qquad (36)$$
$$|\chi_{\varphi_0^{l_1}} \chi_{\varphi_2^{l_2}} \chi_{\varphi_{-2}^{l_3}} + \chi_{\varphi_2^{l_1}} \chi_{\varphi_{-2}^{l_2}} \chi_{\varphi_0^{l_3}} + \chi_{\varphi_{-2}^{l_1}} \chi_{\varphi_0^{l_2}} \chi_{\varphi_2^{l_3}}|^2 +$$
$$|\chi_{\varphi_0^{l_1}} \chi_{\varphi_{-2}^{l_2}} \chi_{\varphi_2^{l_3}} + \chi_{\varphi_{-2}^{l_1}} \chi_{\varphi_2^{l_2}} \chi_{\varphi_0^{l_3}} + \chi_{\varphi_2^{l_1}} \chi_{\varphi_0^{l_2}} \chi_{\varphi_{-2}^{l_3}}|^2.$$

Table 3: Scaling dimensions $h_i$ of the 24 primary fields corresponding to the $c(Z_3^{\times 3})$ and $c(u(1)_6^{\times 3})$ condensation invariants. Here $\{h\}^n$ denotes that there are $n$ distinct primary fields with scaling dimension $h$.

| | $h_i$ |
|---|---|
| $c(Z_3^{\times 3})$ | $0, \{\frac{2}{15}\}^6, \frac{1}{5}, \{\frac{2}{5}\}^3, \{\frac{8}{15}\}^2, \{\frac{11}{15}\}^6, \{\frac{4}{5}\}^3, \{\frac{4}{3}\}^2$ |
| $c(u(1)_6^{\times 3})$ | $0, \{\frac{1}{6}\}^6, \frac{1}{4}, \{\frac{5}{12}\}^6, \{\frac{1}{2}\}^3, \{\frac{2}{3}\}^2, \{\frac{3}{4}\}^3, \{\frac{11}{12}\}^2$ |

This partition function corresponds to the diagonal invariant of a $c = 12/5$ CFT that has 24 primary fields with scaling dimensions given in Table 3. To our knowledge this CFT has not been considered before in literature and we call it here $c(Z_3^{\times 3})$.

Using the properties (23), we find that $Z_{Z_3^{\times 3}}^{\text{c}}$ can be written as the linear combination

$$Z_{Z_3^{\times 3}}^{\text{c}} = \sum_{Q_1, Q_2, Q_3 = 0,1,2} Z_{Q_1}^{-Q_3} Z_{Q_2}^{-Q_3} Z_{Q_3}^{Q_1 + Q_2}. \tag{37}$$

Thus the boundary condition of one of the chains should depend on the product of the sectors of the other two chains, while the boundary conditions of the other two chains should depend on the sector of this chain only. This can be achieved by adding to (35) the boundary coupling

$$H_B = -(\mathcal{P}_3^\dagger - \mathbf{1})X_{L-2}X_1^\dagger - (\mathcal{P}_3^\dagger - \mathbf{1})X_{L-1}X_2^\dagger - (\mathcal{P}_1\mathcal{P}_2 - \mathbf{1})X_L X_3^\dagger + \text{h.c.}, \tag{38}$$

where $\mathcal{P}_n = \prod_{j=0}^{L/3-1} Z_{3j+n}$, for $n = 1, 2, 3$. To write down a local model for the condensation invariant, we employ again canonical duality transformations that now take the form (shown here for a chain of nine sites, the generalization to larger chains is straightforward)

$$Z_1 = \tilde{X}_1^\dagger \tilde{X}_2, \qquad Z_2 = \tilde{X}_2^\dagger \tilde{X}_3, \qquad Z_3 = \tilde{Z}_1 \tilde{Z}_2 \tilde{Z}_3,$$
$$Z_4 = \tilde{X}_4^\dagger \tilde{X}_5, \qquad Z_5 = \tilde{X}_5^\dagger \tilde{X}_6, \qquad Z_6 = \tilde{Z}_4 \tilde{Z}_5 \tilde{Z}_6,$$
$$Z_7 = \tilde{X}_7^\dagger \tilde{X}_8, \qquad Z_8 = \tilde{X}_8^\dagger \tilde{X}_9, \qquad Z_9 = \tilde{Z}_7 \tilde{Z}_8 \tilde{Z}_9,$$

$$X_1 = \tilde{Z}_1, \qquad X_2 = \tilde{Z}_1 \tilde{Z}_2, \qquad X_3 = \tilde{X}_1 \mathcal{P}_1 \mathcal{P}_2,$$
$$X_4 = Z_3 \tilde{Z}_4, \qquad X_5 = Z_3 \tilde{Z}_4 \tilde{Z}_5, \qquad X_6 = Z_1^\dagger Z_2^\dagger \tilde{X}_4 \mathcal{P}_1 \mathcal{P}_2,$$
$$X_7 = Z_3 Z_6 \tilde{Z}_7, \qquad X_8 = Z_3 Z_6 \tilde{Z}_7 \tilde{Z}_8, \qquad X_9 = Z_1^\dagger Z_2^\dagger Z_4^\dagger Z_5^\dagger \tilde{X}_7 \mathcal{P}_2 \mathcal{P}_1.$$

In terms of the dual operators, one finds the translationally invariant chain

$$H_{c(Z_3^{\times 3})} = H_{Z_3^{\times 3}} + H_B = -\sum_{i=1}^L (\tilde{Z}_i \tilde{Z}_{i+1} \tilde{Z}_{i+2} + \text{h.c.}) + (\tilde{X}_i \tilde{X}_{i+1}^\dagger + \text{h.c.}). \tag{39}$$

**The condensation invariant of $u(1)_6^{\times 3}$ CFT**

Switching the sign of the Hamiltonian (35), i.e. considering $-H_{Z_3^{\times 3}}$, gives rise to $u(1)_6^{\times 3}$ criticality. This product CFT also contains primary fields with integer scaling dimensions suggesting that a condensation invariant exists. This invariant is given by

$$Z_{u(1)_6^{\times 3}}^{\text{c}} = \sum_{s_1 = 0,3} \sum_{s_2 = 0,3} \sum_{s_3 = 0,3} \tag{40}$$

$$|\chi_{\varphi_{0+s_1}} \chi_{\varphi_{0+s_2}} \chi_{\varphi_{0+s_3}} + \chi_{\varphi_{2+s_1}} \chi_{\varphi_{2+s_2}} \chi_{\varphi_{2+s_3}} + \chi_{\varphi_{-2+s_1}} \chi_{\varphi_{-2+s_2}} \chi_{\varphi_{-2+s_3}}|^2 +$$

$$|\chi_{\varphi_{0+s_1}} \chi_{\varphi_{2+s_2}} \chi_{\varphi_{-2+s_3}} + \chi_{\varphi_{2+s_1}} \chi_{\varphi_{-2+s_2}} \chi_{\varphi_{0+s_3}} + \chi_{\varphi_{-2+s_1}} \chi_{\varphi_{0+s_2}} \chi_{\varphi_{2+s_3}}|^2 +$$

$$|\chi_{\varphi_{0+s_1}} \chi_{\varphi_{-2+s_2}} \chi_{\varphi_{2+s_3}} + \chi_{\varphi_{-2+s_1}} \chi_{\varphi_{2+s_2}} \chi_{\varphi_{0+s_3}} + \chi_{\varphi_{2+s_1}} \chi_{\varphi_{0+s_2}} \chi_{\varphi_{-2+s_3}}|^2,$$

which corresponds to the diagonal invariant of a $c = 3$ CFT that contains 24 primary fields with scaling dimensions given in Table 3. To construct the microscopic model, we can again follow precisely the steps as in the case of the condensation invariant for $Z_3^{\times 3}$. Thus, the microscopic model for the condensation invariant partition function $Z^c_{u(1)_2^{\times 3}}$ is simply given by

$$H_{c(u(1)_6^{\times 3})} = -H_{c(Z_3^{\times 3})}.$$

## 4.4 The permutation invariants of $Z_3^{\times 3}$ and $u(1)_6^{\times 3}$ CFTs

Apart from the condensation invariant, the $Z_3^{\times 3}$ theory also allows for a non-trivial permutation invariant, which is similar to the permutation invariant of the $Z_3^{\times 2}$ theory. To be explicit, in this case it is given by

$$Z^\pi_{Z_3^{\times 3}} = \sum_{l_i = 0,2} \sum_{m_i = -2,0,2} \chi_{\varphi_{m_1}^{l_1}} \chi_{\varphi_{m_2}^{l_2}} \chi_{\varphi_{m_3}^{l_3}} \bar{\chi}_{\varphi_{m_2}^{l_1}} \bar{\chi}_{\varphi_{m_3}^{l_2}} \bar{\chi}_{\varphi_{m_1}^{l_3}}, \tag{41}$$

which can be expressed as the linear combination

$$Z^\pi_{Z_3^{\times 3}} = \sum_{Q_1,Q_2,Q_3 = 0,1,2} Z_{Q_1}^{-Q_2-Q_3} Z_{Q_2}^{Q_1-Q_3} Z_{Q_3}^{Q_1+Q_2}. \tag{42}$$

This coupling can be implemented in the three decoupled 3-state Potts models by introducing the boundary term

$$H^B_{\pi(Z_3^{\times 3})} = -(\mathcal{P}_2^\dagger \mathcal{P}_3^\dagger - \mathbf{1}) X_{L-2} X_1^\dagger - (\mathcal{P}_1 \mathcal{P}_3^\dagger - \mathbf{1}) X_{L-1} X_2^\dagger - (\mathcal{P}_1 \mathcal{P}_2 - \mathbf{1}) X_L X_3^\dagger + \text{h.c.}, \tag{43}$$

where as above $\mathcal{P}_n = \prod_{j=0}^{L/3-1} Z_{3j+n}$. To rewrite this Hamiltonian in an translationally invariant form, one can use the following canonical transformation (given here again for nine sites for clarity)

$$
\begin{aligned}
Z_1 &= \tilde{X}_1^\dagger \tilde{X}_2, & Z_2 &= \tilde{X}_2^\dagger \tilde{X}_3, & Z_3 &= \tilde{Z}_1 \tilde{Y}_2 \tilde{Z}_3, \\
Z_4 &= \tilde{X}_4^\dagger \tilde{X}_5, & Z_5 &= \tilde{X}_5^\dagger \tilde{X}_6, & Z_6 &= \tilde{Z}_4 \tilde{Y}_5 \tilde{Z}_6, \\
Z_7 &= \tilde{X}_7^\dagger \tilde{X}_8, & Z_8 &= \tilde{X}_8^\dagger \tilde{X}_9, & Z_9 &= \tilde{Z}_7 \tilde{Y}_8 \tilde{Z}_9,
\end{aligned}
$$

$$
\begin{aligned}
X_1 &= \tilde{Z}_1, & X_2 &= \tilde{Y}_1 \tilde{Z}_2 \mathcal{P}_1, & X_3 &= \tilde{X}_1 \mathcal{P}_1 \mathcal{P}_2, \\
X_4 &= Z_2 Z_3 \tilde{Z}_4, & X_5 &= Z_1^\dagger Z_3 \tilde{Y}_4 \tilde{Z}_5 \mathcal{P}_1, & X_6 &= Z_1^\dagger Z_2^\dagger \tilde{X}_4 \mathcal{P}_1 \mathcal{P}_2, \\
X_7 &= Z_2 Z_3 Z_5 Z_6 \tilde{Z}_7, & X_8 &= Z_1^\dagger Z_3 Z_4^\dagger Z_6 \tilde{Y}_7 \tilde{Z}_8 \mathcal{P}_1, & X_9 &= Z_1^\dagger Z_2^\dagger Z_4^\dagger Z_5^\dagger \tilde{X}_7 \mathcal{P}_1 \mathcal{P}_2,
\end{aligned}
$$

where $Y_i = Z_i X_i$. In terms of the dual variables one obtains the local and translationally invariant Hamiltonian

$$H_{\pi(Z_3^{\times 3})} = H_{Z_3^{\times 3}} + H^B_{\pi(Z_3^{\times 3})} = -\sum_{i=1}^{L} (\tilde{Z}_i \tilde{Y}_{i+1} \tilde{Z}_{i+2} + \text{h.c.}) + (\tilde{X}_i \tilde{X}_{i+1}^\dagger + \text{h.c.}). \tag{44}$$

**The permutation invariant of $u(1)_6^{\times 3}$ CFT**

The $u(1)_6^{\times 3}$ CFT admits also a permutation invariant corresponding to the partition function

$$Z^\pi_{u(1)_6^{\times 3}} = \sum_{m_i = -2}^{3} \chi_{\varphi_{m_1}} \chi_{\varphi_{m_2}} \chi_{\varphi_{m_3}} \bar{\chi}_{\varphi_{3m_1+2m_2}} \bar{\chi}_{\varphi_{3m_2+2m_3}} \bar{\chi}_{\varphi_{3m_3+2m_1}}. \tag{45}$$

The analysis goes again through identically and thus the microscopic model realizing this invariant is given by $H_{\pi(u(1)_6^{\times 3})} = -H_{\pi(Z_3^{\times 3})}$. This applies also to the generic case of $\pi(u(1)_6^{\times n})$, which means that a permutation invariant can be realized in perturbed cluster clock models $H_{\pi(u(1)_6^{\times n})} = -H_{\pi(Z_3^{\times n})}$.

## 4.5 Condensation and permutation invariants for $Z_3^{\times N}$ and $u(1)_6^{\times N}$ CFTs

Finally, we consider a few cases that generalize our construction of microscopic critical models to $Z_3^{\times N}$ and $u(1)_6^{\times N}$ CFTs for generic $N$. As $N$ increases, the number of possible non-diagonal modular invariants grows quickly. We therefore limit ourselves to two cases, that are straightforward generalizations of the condensation and permutations invariants, as described in Sections 4.3 and 4.4.

The starting point is the Hamiltonian

$$H_{Z_3^{\times N}} = -\sum_{i=1}^{L}(X_i X_{i+N}^\dagger + Z_i + \text{h.c.}), \tag{46}$$

that describes $N$ decoupled 3-state Potts models. As evaluating the modular invariants (8) explicitly become intractable for generic $N$, we instead consider generalizations of the boundary terms (38) and (43) as ansatzes for non-diagonal modular invariants. These are given by

$$H_{c(Z_3^{\times N})}^B = -\sum_{n=1}^{N-1}\left((\mathcal{P}_N^\dagger - \mathbf{1})X_{L-N+n}X_n^\dagger + \text{h.c.}\right) - \left((\prod_{n=1}^{N-1}\mathcal{P}_n - \mathbf{1})X_L X_N^\dagger + \text{h.c.}\right) \tag{47}$$

and

$$H_{\pi(Z_3^{\times N})}^B = -\sum_{n=1}^{N}(\mathcal{P}_1^\dagger \cdots \mathcal{P}_{n-1}^\dagger \mathcal{P}_{n+1} \cdots \mathcal{P}_N - \mathbf{1})X_{L-N+n}X_n^\dagger + \text{h.c.}\,, \tag{48}$$

respectively, where now $\mathcal{P}_n = \prod_{j=0}^{L/N-1} Z_{Nj+n}$, for $n = 1, 2, \ldots N$. Similar to the condensation boundary term (38) for three chains, $H_{c(Z_3^{\times N})}^B$ couples the chains such that the boundary condition of the $N^{\text{th}}$ chain depends on the symmetry sector of the first $N-1$ chains, while the boundary conditions of the first chains $N-1$ chains depend only on the symmetry sector of the $N^{\text{th}}$ chain. Likewise, $H_{\pi(Z_3^{\times N})}^B$ couples the chains such that the boundary condition of each chain depends on the symmetry sectors of all the other chains, in analogy to the permutation boundary term (43).

The Hamiltonian $H_{Z_3^{\times N}} + H_{c(Z_3^{\times N})}^B$ can be transformed into a translationally invariant and local form by means of a generalization of the canonical transformations given in Sec. 4.3. The resulting Hamiltonian is given by

$$H_{Z_3^{\times N}} + H_{c(Z_3^{\times N})}^B = -\sum_{i=1}^{L}(\tilde{Z}_i \tilde{Z}_{i+1} \cdots \tilde{Z}_{i+N-1} + \text{h.c.}) + (\tilde{X}_i \tilde{X}_{i+1}^\dagger + \text{h.c.})\,, \tag{49}$$

which is a generalization of the Hamiltonian (39) for the condensation invariant $c(Z_3^{\times 3})$. We verified that this model is again critical and described by a non-diagonal modular invariant for all $N$. Interestingly, we find that the type of invariant depends on $N$. If $N$ is a multiple of three, the invariant is equal to a condensation invariant of the CFT consisting of the direct product of $N$ $Z_3$ parafermion CFTs. This condensation invariant is a straightforward generalization of the condensation invariant (36). The field that condenses is $(\varphi_2^0, \varphi_2^0, \ldots, \varphi_2^0)$, which has scaling dimension $h = 2N/3$ and which is thus a boson for $N \bmod 3 = 0$. If $N$ is not a multiple of three, the invariant is a permutation invariant obtained via the permutation (71) discussed in Appendix A.

By changing the overall sign of the Hamiltonian (49), we obtain a model that realizes a non-diagonal modular invariant of the $u(1)_6^{\times N}$ CFT. Again, for $N$ a multiple of three, the invariant is a straightforward generalization of the condensation invariant (40) for three 3-state Potts chains. The field that condenses is $(\varphi_2, \varphi_2, \ldots, \varphi_2)$, which has the scaling dimension

$h = N/3$. In the other cases, the invariant is equal to the permutation invariant obtained via the permutation (66) discussed in Appendix A.

Like above, the Hamiltonian $H_{Z_3^{\times N}} + H^B_{\pi(Z_3^{\times N})}$ can be transformed into a translationally invariant and local form by means of a generalization of the canonical transformation given in Sec. 4.4. The resulting Hamiltonian is given by

$$H_{Z_3^{\times N}} + H^B_{\pi(Z_3^{\times N})} = -\sum_{i=1}^{L}(\tilde{Z}_i \tilde{Y}_{i+1} \cdots \tilde{Y}_{i+N-2} \tilde{Z}_{i+N-1} + \text{h.c.}) + (\tilde{X}_i \tilde{X}_{i+1}^{\dagger} + \text{h.c.}), \qquad (50)$$

which is now a generalization of the Hamiltonian (44) for the permutation invariant $\pi(Z_3^{\times 3})$. We have again verified that this Hamiltonian is critical and described by a non-diagonal modular invariant. However, unlike (49), the corresponding invariant is now always a permutation invariant for all values of $N$. The permutation describing this invariant is given by (70), as discussed in Appendix A. Finally, changing the sign of (50) gives a critical model described by permutation invariants of $u(1)_6^{\times N}$ CFT for all values of $N$. The corresponding permutation is given by (65) in Appendix A.

There are intriguing parallels of the terms $\tilde{Z}_i \tilde{Y}_{i+1} \cdots \tilde{Y}_{i+n-2} \tilde{Z}_{i+n-1}$ to those appearing in generalized spin-1/2 cluster models [6]. To be precise, were the $\tilde{Z}_i$ and $\tilde{X}_i$ operators replaced by Pauli matrices, then (50) would realize a hierarchy of critical models described by all $so(N)_1$ CFTs (that correspond to condensation invariants of Ising$^{\times N}$ CFTs [18]). Whether this similarity is accidental, or whether there is some deeper connection, remains an open question. This similarity suggests also the existence of a clock cluster phase. Were the perturbations $\tilde{X}_i \tilde{X}_{i+1}^{\dagger} + \text{h.c.}$ small compared to the cluster terms $\tilde{Z}_i \tilde{Y}_{i+1} \cdots \tilde{Y}_{i+n-2} \tilde{Z}_{i+n-1} + \text{h.c.}$ (named as such due them all commuting with each other), one expects a gapped, possibly symmetry-protected topological phase with some number of parafermion edge states [25]. To our understanding such models have not been explored in the literature and it would be interesting to investigate whether they exhibit interesting entanglement properties akin to spin-1/2 cluster states [51].

# 5  Beyond 3-state Potts models: Non-diagonal invariants of $Z_k^{\times 2}$ CFTs

A natural way to generalize our construction is to go from 3-state Potts models to clock models with $Z_k$ symmetry [29]. Such models are obtained by redefining the clock operators $Z$ and $X$ to describe $k$-state systems. Instead of (2), we demand that they now satisfy $Z^k = X^k = \mathbf{1}$, $ZX = \omega XZ$, $Z^{k-1} = Z^{\dagger} = Z^{-1}$ and $X^{k-1} = X^{\dagger} = X^{-1}$, where $\omega = e^{2\pi i/k}$. The phase diagrams of $k$-state Potts models were already considered in [29]. It is known that when the relative magnitudes of the different powers of the clock operators are tuned suitably, they exhibit critical points described described by $Z_k$ parafermion CFTs, that we discussed in Section 3. For every $k$, the critical microscopic Hamiltonians take the form [29]

$$H_{Z_k} = -\sum_{i} \sum_{p=1}^{k-1} \frac{1}{\sin(p\pi/k)} \big((Z_i^p) + (X_i^p X_{i+1}^{-p})\big), \qquad (51)$$

which is Hermitian due to the sum over $p$.

While the $k$-state models exhibit also other critical points, we focus on this case to show that our construction is not limited to $Z_3$ parafermions only. Even with the restriction to the Hamiltonian (51), there are many ways in which we can use our construction to create further microscopic chains with particular CFTs. One option is to consider $k$ decoupled chains and condense the boson that is present in the spectrum (see Section 3 for the field content of $Z_k$

parafermion CFTs). However, the number of primary fields grows very quickly upon increasing $k$. Hence, we focus on two decoupled $k$-state clock chains, described by the diagonal invariant of $Z_k^{\times 2}$ CFT, and look for a generalization of the permutation invariant discussed in Section 4.1. As in the preceding Section 4.5, solving for all the modular invariants for a generic $k$ becomes intractable. Thus we again make an ansatz for a suitable boundary term by generalizing the 3-state construction and verify that the resulting model is indeed critical and described by a non-diagonal modular invariant.

We start with the Hamiltonian, which has the $Z_k^{\times 2}$ CFT describing its critical point,

$$H_{Z_k^{\times 2}} = -\sum_i \sum_{p=1}^{k-1} \frac{1}{\sin(p\pi/k)}\big((Z_i^p) + (X_i^p X_{i+2}^{-p})\big). \tag{52}$$

Motivated by (29), we introduce the boundary term

$$H_B = -\sum_{p=1}^{k-1} \frac{1}{\sin(p\pi/k)}\Big(X_{L-1}^p X_1^{-p}\big(\mathcal{P}_2^p - \mathbf{1}\big) + X_L^p X_2^{-p}\big(\mathcal{P}_1^{-p} - \mathbf{1}\big)\Big), \tag{53}$$

where $\mathcal{P}_n = \prod_{j=0}^{L/2-1} Z_{2j+n}$ as before. After a canonical transformation to dual $k$-state clock variables, namely Eq. (30) for $k$-state clock variables, one finds that the Hamiltonian $H_{Z_k^{\times 2}}^{c/\pi} = H_{Z_k^{\times 2}} + H_B$ takes the following local and translational invariant form

$$H_{Z_k^{\times 2}}^{c/\pi} = -\sum_i \sum_{p=1}^{k-1} \frac{1}{\sin(p\pi/k)}\big(\tilde{Z}_i^p \tilde{Z}_{i+1}^p + \tilde{X}_i^p \tilde{X}_{i+1}^{-p}\big). \tag{54}$$

We verified that this Hamiltonian is indeed critical and described by a non-diagonal modular invariant. Interestingly, we observe that for $k$ odd, the new invariant is a permutation invariant of the $Z_k^{\times 2}$ CFT, while for $k$ even, the new invariant is a condensation invariant instead. This behaviour is consistent with the behaviour observed for $k = 2$ (condensation in Ising$^{\times 2}$ [17]) and $k = 3$ (Section 4.1). To conclude this section, we give these non-diagonal invariants of $Z_k^{\times 2}$ CFT explicitly.

## 5.1 Permutation invariants for odd $k$

When $k$ is odd, the modular invariant realized by the Hamiltonian Eq. (54) is a permutation invariant, which is a direct generalization of the $k = 3$ case discussed in Section 3. The chiral fields of the $Z_k^{\times 2}$ CFT can be labeled as $(\varphi_{m_1}^{l_1}, \varphi_{m_2}^{l_2})$, where $l_1, l_2 = 0, 2, \dots, k-1$ and $m_1, m_2 = -k+1, -k+3, \dots, k-3, k-1$. The permutation that describes the invariant is given by

$$\pi(\varphi_{m_1}^{l_1}, \varphi_{m_2}^{l_2}) = (\varphi_{m_2}^{l_1}, \varphi_{-m_1}^{l_2}), \tag{55}$$

which gives rise to the permutation modular invariant

$$Z_{Z_k^{\times 2}}^{\pi} = \sum_{\substack{l_1, l_2 = 0, 2, \dots k-1 \\ m_1, m_2 = -k+1, -k+3, \dots, k-3, k-1}} \chi_{(\varphi_{m_1}^{l_1}\varphi_{m_2}^{l_2})}\bar{\chi}_{(\varphi_{m_2}^{l_1}\varphi_{-m_1}^{l_2})}. \tag{56}$$

This modular invariant partition function clearly has a distinct spectrum from the diagonal modular invariant.

## 5.2 Condensation invariants for even $k$

In the case of $k$ even, the chiral fields of the $Z_k^{\times 2}$ CFT can be labeled as $(\varphi_{m_1}^{l_1}, \varphi_{m_2}^{l_2})$, where now $l_1, l_2 = 0, 1, \ldots, k/2$. In addition, $m_1, m_2$ must have the same parity as $l_1, l_2$, and lie in the range $0, 1, \ldots, 2k-1$ for $l_i < k/2$ and $0, 1, \ldots, k-1$ for $l_i = k/2$. The modular invariant realized by the microscopic model (54) corresponds to a condensation invariant that is obtained by condensing the boson $(\varphi_k^0, \varphi_k^0)$ with scaling dimension $h = k/2$, followed by a permutation of the fields. The closed form of the modular invariant partition function depends on the parity of $k/2$.

Explicitly, for $k = 0 \bmod 4$, this condensation invariant takes the following form (we remind the reader that the $m$ label is defined modulo $2k$)

$$
\begin{aligned}
Z^c_{Z_k^{\times 2}} = &\sum_{\substack{l_1, l_2 = 1,3,\ldots,k/2-1 \\ m_1 = 1,3,\ldots,k-1 \\ m_2 = 1,3,\ldots,2k-1}} |\chi_{(\varphi_{m_1}^{l_1}, \varphi_{m_2}^{l_2})} + \chi_{(\varphi_{m_1+k}^{l_1}, \varphi_{m_2+k}^{l_2})}|^2 + \\
&\sum_{\substack{l_1, l_2 = 0,2,\ldots,k/2-2 \\ m_1 = 0,2,\ldots,k-2 \\ m_2 = 0,2,\ldots,2k-2}} |\chi_{(\varphi_{m_1}^{l_1}, \varphi_{m_2}^{l_2})} + \chi_{(\varphi_{m_1+k}^{l_1}, \varphi_{m_2+k}^{l_2})}|^2 + \\
&\sum_{\substack{l_1 = k/2, l_2 = 0,2,\ldots,k/2-2 \\ m_1, m_2 = 0,2,\ldots,k-2}} |\chi_{(\varphi_{m_1}^{l_1}, \varphi_{m_2}^{l_2})} + \chi_{(\varphi_{m_1+k}^{l_1}, \varphi_{m_2+k}^{l_2})}|^2 + \\
&\sum_{\substack{l_1 = 0,2,\ldots,k/2-2, l_2 = k/2 \\ m_1, m_2 = 0,2,\ldots,k-2}} |\chi_{(\varphi_{m_1}^{l_1}, \varphi_{m_2}^{l_2})} + \chi_{(\varphi_{m_1+k}^{l_1}, \varphi_{m_2+k}^{l_2})}|^2 + \\
&\sum_{\substack{l_1 = l_2 = k/2 \\ m_1, m_2 = 0,2,\ldots,k-2}} 2 |\chi_{(\varphi_{m_1}^{l_1}, \varphi_{m_2}^{l_2})}|^2 .
\end{aligned} \tag{57}
$$

On the other hand, when $k = 2 \bmod 4$, the largest possible values for $l_1, l_2$ is $k/2$, which is now odd. The explicit form of condensation invariant is slightly different,

$$
\begin{aligned}
Z^c_{Z_k^{\times 2}} = &\sum_{\substack{l_1, l_2 = 0,2,\ldots,k/2-1 \\ m_1 = 0,2,\ldots,k-2 \\ m_2 = 0,2,\ldots,2k-2}} |\chi_{(\varphi_{m_1}^{l_1}, \varphi_{m_2}^{l_2})} + \chi_{(\varphi_{m_1+k}^{l_1}, \varphi_{m_2+k}^{l_2})}|^2 + \\
&\sum_{\substack{l_1, l_2 = 1,3,\ldots,k/2-2 \\ m_1 = 1,3,\ldots,k-1 \\ m_2 = 1,3,\ldots,2k-1}} |\chi_{(\varphi_{m_1}^{l_1}, \varphi_{m_2}^{l_2})} + \chi_{(\varphi_{m_1+k}^{l_1}, \varphi_{m_2+k}^{l_2})}|^2 + \\
&\sum_{\substack{l_1 = k/2, l_2 = 1,3,\ldots,k/2-2 \\ m_1, m_2 = 1,3,\ldots,k-1}} |\chi_{(\varphi_{m_1}^{l_1}, \varphi_{m_2}^{l_2})} + \chi_{(\varphi_{m_1+k}^{l_1}, \varphi_{m_2+k}^{l_2})}|^2 + \\
&\sum_{\substack{l_1 = 1,3,\ldots,k/2-2, l_2 = k/2 \\ m_1, m_2 = 1,3,\ldots,k-1}} |\chi_{(\varphi_{m_1}^{l_1}, \varphi_{m_2}^{l_2})} + \chi_{(\varphi_{m_1+k}^{l_1}, \varphi_{m_2+k}^{l_2})}|^2 + \\
&\sum_{\substack{l_1 = l_2 = k/2 \\ m_1, m_2 = 0,2,\ldots,k-1}} 2 |\chi_{(\varphi_{m_1}^{l_1}, \varphi_{m_2}^{l_2})}|^2 .
\end{aligned} \tag{58}
$$

For both cases the corresponding CFT has $(k/2)^2((k/2)^2 + k/2 + 2)$ primary fields and central charge $c = \frac{4(k-1)}{(k+2)}$.

To obtain the modular invariant realized by the Hamiltonian (54), denoted by $Z^{\pi \circ c}_{Z_k^{\times 2}}$, one has to perform a further permutation of the anti-chiral fields on these condensation invariants.

The needed permutation is the same as the permutation (55) used to construct the permutation invariants for $k$ odd. Applying it to the partition functions above is equivalent to replacing all the anti-chiral characters $\bar{\chi}_{(\varphi_{m_1}^{l_1}, \varphi_{m_2}^{l_2})}$ by $\bar{\chi}_{(\varphi_{m_2}^{l_1}, \varphi_{-m_1}^{l_2})}$. This does not change the central charge or the number of primary fields and the resulting modular invariants $Z_{Z_k^{\times 2}}^{\pi \circ c}$ completely specify criticality of the Hamiltonian (54) for all $k$.

## 6 Conclusion

We have constructed several new microscopic clock-type models that by construction exhibit exotic critical points described by CFTs not considered previously in the literature. Some of these correspond to non-trivial permutation invariant partition functions, which shows explicitly that such modular invariants allowed by theory also have realizations in local and translational invariant models. While not limited to them, we focused on generalizations of 3-state Potts models. When these are mapped into parafermion chains with the Fradkin-Kadanoff transformation, our models correspond to parafermion chains with many-body terms. Thus our work addresses also the open question of phase transitions that such many-body terms can drive if they were to appear.

First and foremost, our main result is to demonstrate that exotic CFTs can appear in local and translational invariant models and to construct representative models for each, as summarized in Table 1. The physical realization of any Potts model is challenging, but proposals exist suggesting that domain walls in Abelian fractional quantum Hall edges could realize the parafermion modes [22, 23], whose hybridization can result in parafermion chains that are unitarily equivalent to Potts models [14]. If the hybridization were also to give rise to many-body terms, then our critical models could also be realized in this setting. While speculative, if they were, following the same principles to construct a 2D topologically ordered phase from coupled critical 3-state Potts models with $Z_3$ parafermion criticality, our models could serve as building blocks for even more exotic states of topological 2D matter [14, 15].

As we have shown by considering also critical generalizations of $k$-state Potts models, our method is versatile and lends itself for construction of local models for ever more exotic CFTs. While this is a rather academic exercise with the models increasing in complexity, ideally one would like to have local microscopic realizations for all modular invariants of all CFTs. Admittedly, this is a formidable task as the classification of all modular invariant partition functions is an open question. Still, progress towards this goal can be made with methods of presented here. To go beyond 3-state Potts models, we considered a few examples of models that can be constructed by coupling together two $k$-state clock models. A particularly interesting case to consider would be to coupling together two such models with opposite sign, generalizing the case of Sec. 4.2. Using the results in [49], i.e. that the central charge for the antiferromagnetic $Z_k$ parafermion chains is $c = 1$, one would expect that the resulting critical point is described by the $su(2)_k$ CFT[3]. These in turn can be used to describe minimal models via the coset construction [48], thus possibly enabling a different construction of microscopic realizations for all CFTs in this important class.

While our focus has been on the criticality of the models constructed, it would also be interesting to understand how the criticality relates to the symmetry protected topological phases in the vicinity of the critical points [8, 25]. The models with commuting cluster like terms (50) should also be investigated. In spin-1/2 models such terms give rise to entangled ground states that are universal resources for quantum computation [51]. It would be interesting to

---

[3]For $k = 5, 7$, it has been shown in [50] that the CFT describing the antiferromagnetic $Z_k$ chains is $u(1)_{2k}$, as required.

investigate what, if any, new interesting phenomena appear in the clock model counterparts of cluster states.

Finally, we would like to point out that a generalization of the 3-state Potts model, that is slightly different from the one considered in Sec. 4.1, Eq. (31) was studied in [52]. That model can be obtained simply by exchanging the $Z_i Z_{i+1}$ term in Eq. (31) by $Z_i Z_{i+1}^\dagger$. This cosmetically small change leads to rather different behaviour of the model, for which it turned out hard to fully characterize the critical point. It would be interesting to investigate the relations between these models, if any.

# Acknowledgements

E.A. would like to thank Jérôme Dubail, Paul Fendley and Roger Mong for interesting discussions.

**Funding information**  V.L. is supported by the Dahlem Research School POINT fellowship program. E.A. is supported, in part, by the Swedish research council.

# A  Fusion rule symmetries of CFTs

In this appendix we discuss the possible permutations of the primary fields in the $Z_3$ parafermion and $u(1)_6$ CFTs that leave their fusion rules invariant. Such fusion rule symmetries enable to predict possible non-diagonal modular invariants that may exist in product theories. To this end we treat $Z_k$ parafermion CFTs as the cosets $Z_k \simeq su(2)_k/u(1)_{2k}$.

The $su(2)_k$ CFT (with $k$ a positive integer) has $k+1$ primary fields that we denote by $\varphi^l$, with $l = 0, 1, \ldots, k$. These satisfy the fusion rules

$$\varphi^{l_1} \times \varphi^{l_2} = \varphi^{|l_1 - l_2|} + \varphi^{|l_1 - l_2| + 1} + \cdots + \varphi^{\max(l_1 + l_2, 2k - l_1 - l_2)} . \tag{59}$$

On the other hand, $u(1)_{2k}$ contains $2k$ primary fields that we denote by $\varphi_m$, where $m$ is an integer defined modulo $2k$. We adopt a convention that $m$ lies in the range $-k < m \le k$. These primary fields obey the fusion rules

$$\varphi_{m_1} \times \varphi_{m_2} = \varphi_{m_1 + m_2} . \tag{60}$$

Treating the $Z_k$ parafermion CFT as the coset $su(2)_k/u(1)_{2k}$ means that its primary fields are labeled by the labels of the $su(2)_k$ and $u(1)_{2k}$ theories, namely $\varphi_m^l$. The coset means that the labels are no longer independent though, but subject to the constraint $l + m = 0 \bmod 2$ and to the field identification $\varphi_m^l \equiv \varphi_{m+k}^{k-l}$. With these constraints, the fusion rules follow directly from the fusion rules of the $su(2)_k$ and $u(1)_{2k}$ theories,

$$\varphi_{m_1}^{l_1} \times \varphi_{m_2}^{l_2} = \varphi_{m_1 + m_2}^{|l_1 - l_2|} + \varphi_{m_1 + m_2}^{|l_1 - l_2| + 1} + \cdots + \varphi_{m_1 + m_2}^{\max(l_1 + l_2, 2k - l_1 - l_2)} . \tag{61}$$

For direct product theories the fusion rules are simply direct products of the fusion rules of the constituent fields. For instance, a product of two $Z_3$ parafermion CFTs has 36 primary fields $(\varphi_{m_1}^{l_1}, \varphi_{m_2}^{l_2})$. They satisfy the fusion rules

$$(\varphi_{m_1}^{l_1}, \varphi_{m_2}^{l_2}) \times (\varphi_{m_3}^{l_3}, \varphi_{m_4}^{l_4}) = (\varphi_{m_1}^{l_1} \times \varphi_{m_3}^{l_3}, \varphi_{m_2}^{l_2} \times \varphi_{m_4}^{l_4}), \tag{62}$$

where the right hand side is given by all possible product fields compatible with the $Z_3$ fusion rules.

## A.1 Symmetries of $u(1)_6^{\times n}$ fusion rules

Let us consider first the symmetries of $u(1)_6$ fusion rules and product theories of them. For a single $u(1)_6$ CFT, the only possible permutation invariant is $\varphi_m \to \varphi_{-m}$ due to the cyclic nature of the fusion rules. This permutation does not lead to a distinct partition function. For a product of two such CFTs, there are more alternatives. All the permutations

$$m_1 \leftrightarrow m_2, \qquad m_1 \to -m_1 \quad \text{or} \quad m_2 \to -m_2, \tag{63}$$

that either swap the two theories or apply the single theory permutations, leave the fusion rules invariant, but again do not change the partition function. On the other hand, there is also a further allowed permutation given by

$$(\varphi_{m_1}, \varphi_{m_2}) \to (\varphi_{m_1+2(m_1+m_2)}, \varphi_{m_2+2(m_1+m_2)}), \tag{64}$$

which results in the permutation invariant partition function discussed in Section 4.1.

The fusion rules of any product theories of three or more $u(1)_6$ CFTs are invariant when the copies are permuted in an arbitrary way, or under inversion symmetries (63). These permutations do not lead to a change in the partition function. When the number of copies increases, the number of non-trivial permutation invariants grows. We do not give an exhaustive list, but mention two types of non-trivial permutation invariants, relevant for the models considered in this paper.

For an arbitrary number $N$ of copies, one obtains a non-trivial permutation invariant by sending

$$(\varphi_{m_1}, \varphi_{m_2}, \dots, \varphi_{m_{N-1}}, \varphi_{m_N}) \to (\varphi_{3m_1+2m_2}, \varphi_{3m_2+2m_3}, \dots, \varphi_{3m_{N-1}+2m_N}, \varphi_{3m_N+2m_1}). \tag{65}$$

Another example is the following permutation, in the case where the number of copies $N$ is not a multiple of three,

$$(\varphi_{m_1}, \varphi_{m_2}, \dots, \varphi_{m_N}) \to (\varphi_{m_1+\alpha m}, \varphi_{m_2+\alpha m}, \dots, \varphi_{m_N+\alpha m}), \tag{66}$$

where $m = \sum_i m_i$ and $\alpha = -2$ if $N \bmod 3 = 1$ and $\alpha = 2$ if $N \bmod 3 = 2$.

## A.2 Symmetries of $Z_3^{\times n}$ parafermion fusion rules

We now turn to the permutations of the $Z_3$ parafermionic fields that leave the fusion rules invariant. We choose the labels of the fields of a single $Z_3$ parafermion theory as $\varphi_m^l$, where $l = 0, 2$ and $m = -2, 0, 2$. The only permutation that leaves the fusion rules invariant is $\varphi_m^l \to \varphi_{-m}^l$ that derives from the $u(1)_6$ label inversion symmetry. Because the characters associated with $\varphi_m^l$ and $\varphi_{-m}^l$ are identical, this permutation does not lead to a partition function that differs from the diagonal partition function.

The situation is different if we consider the product of two $Z_3$ parafermion theories, with the 36 primary fields $(\varphi_{m_1}^{l_1}, \varphi_{m_2}^{l_2})$. There are 16 possible permutations that leave the fusion rules invariant. They are formed by combining in different ways the four elementary permutations

$$l_1 \leftrightarrow l_2, \quad m_1 \leftrightarrow m_2, \quad m_1 \to -m_1 \quad \text{and} \quad m_2 \to -m_2. \tag{67}$$

By considering their action on the partition functions, one finds that only the 8 permutations (the signs are independent)

$$(\varphi_{m_1}^{l_1}, \varphi_{m_2}^{l_2}) \to (\varphi_{\pm m_1}^{l_2}, \varphi_{\pm m_2}^{l_1}) \quad \text{or} \quad (\varphi_{\pm m_2}^{l_1}, \varphi_{\pm m_1}^{l_2}) \tag{68}$$

give rise to the permutation invariants discussed in Section (4.1), while all other are equal to the diagonal invariant.

Permutations that leave the fusion rules invariant generalize directly to a product of three $Z_3$ parafermion theories. The fields are now denoted as $(\varphi_{m_1}^{l_1}, \varphi_{m_2}^{l_2}, \varphi_{m_3}^{l_3})$, and all the permutations that leave the fusion rules invariant are given by combinations of permuting the labels $(l_1, l_2, l_3)$ or $(m_1, m_2, m_3)$ and changing the sign for any of the $m_i$. These permutations give rise to three different partition functions: the diagonal invariant, the product of $Z_3$ and the permutation invariant of $Z_3^2$, i.e. the permutation acts only on two of the three theories in the product, as well as the permutation invariant $\pi(Z_3^\times)$ that involves all three theories. This invariant can be constructed by, for instance, sending

$$(\varphi_{m_1}^{l_1}, \varphi_{m_2}^{l_2}, \varphi_{m_3}^{l_3}) \to (\varphi_{m_2}^{l_1}, \varphi_{m_3}^{l_2}, \varphi_{m_1}^{l_3}). \tag{69}$$

The non-trivial permutation invariant of this type easily generalizes to an arbitrary number of copies, by sending

$$(\varphi_{m_1}^{l_1}, \varphi_{m_2}^{l_2}, \ldots, \varphi_{m_{N-1}}^{l_{N-1}}, \varphi_{m_N}^{l_N}) \to (\varphi_{m_2}^{l_1}, \varphi_{m_3}^{l_2}, \ldots, \varphi_{m_N}^{l_{N-1}}, \varphi_{m_1}^{l_N}) . \tag{70}$$

Again, when the number of copies increases, there are more non-trivial permutations that give non-trivial permutation invariants. We give one more example, namely for $N$ copies one has

$$(\varphi_{m_1}^{l_1}, \varphi_{m_2}^{l_2}, \ldots, \varphi_{m_N}^{l_N}) \to (\varphi_{m_1+Nm}^{l_1}, \varphi_{m_2+Nm}^{l_2}, \ldots, \varphi_{m_N+Nm}^{l_N}) , \tag{71}$$

where $m = \sum_i m_i$ as above. We should note that for $N \bmod 3 = 0$, this is not actually a permutation of the fields, instead the fields are send to themselves because of the field identification.

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
