# Peer review of "Quantum criticality in many-body parafermion chains"

_SciPost Physics Core, doi:SciPost Phys. Core 4, 014 (2021)_

## Round 1 · Referee Report · Anonymous (Referee 1) · 2017-11-15

Strengths

1- Thorough examination of modular invariants in Z_3 invariant systems

2- Many explicit formulas make analysis clear

3- Explicit lattice models realising this behaviour

Weaknesses

1- None of it very profound, but rather an incremental extension of previous knowledge

2- Don't explain the relation to the existing CFT literature, especially that on orbifolds

3- No interesting physical applications (potential or actual)

Report

The authors have constructed modular invariants in conformal field theory by taking copies of three-state Potts model (with a brief generalisation to Z_k parafermions), and then either ``condensing anyons'' 0r utilising permutation invariants. They explain their construction clearly, and work out numerous of examples. They show nicely how to relate the various partition functions by taking linear combinations. They also show how to realise the models as quantum ``spin'' chain Hamiltonians (although not as 2d classical models).

I think the paper is worth of publishing in SciPost. It's good to keep developing methods of conformal field theory, especially those influenced by work in other fields, such as the study of anyons. However, I don't think the results have much physical significance; they don't really explain how their allegedly exotic critical points differ in any substantive way from just taking copies of Potts models. They don't provide any examples of physical behaviour being different as a result (I am aware that there is, but they don't discuss it).

Instead, they say trite things like "Usually, when talking about a critical point being described a given CFT, one refers to the diagonal invariant". That's simply not true (perhaps the authors' friends do, but that shouldn't be extrapolated to the rest of the world). The literature certainly gives lie to that fact. As I believe the authors know (they certainly ought to), when we discusses an extended chiral algebra (such as supersymmetry), often the non-diagonal modular invariant becomes diagonal. Literally hundreds of papers were written in the CFT world about these algebras, and it spawned an entire field of mathematicas (fusion categories etc, which the authors barely mention). To give one of many examples, I note that the authors don't even refer to Ginsparg's famous paper on c=1 theories, where he very clearly explains how to do an orbifold to construct various modular invariants from each other, in exactly the same fashion as the authors describe here. Again, literally hundreds of papers were written on orbifolds, so to say that people tend to always talk about diagonal modular invariants.

Thus the authors need to clarify the relation of their results to the existing CFT literature. In particular, they need to explain how it relates to the orbifold construction; I suspect many of their procedures are completely equivalent.

In general, the authors make a number of small comments like this that I believe give the wrong impression. I list them below.

Requested changes

1- As mentioned above, they need to at least briefly discuss extended chiral algebras, and the relation to the pure Virasoro characters to extended characters. They should also be precise the relation of their construction to the orbifold construction.

2- At the beginning of section 2, they omit some key words. The partition function of every 2d CFT ON THE TORUS must be modular invariant.

3- Again at the beginning of section 2, for many properties of a CFT (e.g. the specific heat), it DOES NOT MATTER IF YOU ARE ON THE TORUS, and so the distinction between different modular invariants is not important. And as I said above, the literature (especially in the '80s) is filled with CFT papers discussing other modular invariants. The authors don't mention these, and don't mention any physical properties that are different (other than the excited state spectrum).

4- on p.7, they say condensation makes less fields. That's not necessarily true, since in condensations, you double other fields.

5- when discussing the Z_2 x Z_2 case, they need to mention GInsparg.

6- they should mention that their added terms to the Hamiltonian are conventionally called "twisted" boundary conditions, and are widely studied.

7- at the beginning of sec 3.1, I don't know what "on-site" Z_k symmetry means. If this is on-site, what is off-site?

8- at the beginning of sec 4.2, they mention that the Bethe ansatz solution gives a factor of 2 in the relative Fermi velocities between the two critical points in Potts. That presumably means if you take the Hamiltonian limit of the same classical 3-state Potts model, this relation holds. That's maybe interesting, but why is this relevant here? They're not studying the classical model.

9- they should mention that the "other" critical point for 3-state corresponds to the anti-ferro Potts model. The fact that this is c=1 goes back at least to Saleur, Nucl.Phys. B360 (1991) 219

10- for general k, they shouldn't say they're generalizing the Potts chain. The conventional definition of the Potts chain is that it has S_k permutation symmetry, and it is not critical for k>4. The points they describe are usually called the Z_k parafermion critical points, and this is what their modular invariants are related to.

11- in the conclusion, they mention getting SU(2)_3 by combining ferro and anti-ferro Potts chains, and speculate this may be true for other k. That seems unlikely, unless the antiferromagnetic point for Z_k parafermions is always c=1. (If that's true, the authors should note it -- I don't know offhand the answer myself).

---

## Round 2 · Author Response

Dear editor and referee,

We would like to start by mentioning that due to various reasons, the resubmission of our manuscript did not materialise. We assumed that it wasn't possible anymore to resubmit, but we were contacted, and told that resubmission was still possible. We of course fully understand if the referee and/or editor has changed her/his/their opinion about the manuscript.

We are pleased to see that the points mentioned by the referee under 'Strength' agree with what we think of the main points of the paper. We agree with the referee about the weaknesses of the paper. The results are more straightforward than profound, but we think, and the referee seems to agree, still interesting. It is also true that there are no physical applications, and having such applications would make the paper more interesting. Though not an excuse, our paper is not alone in this respect. There is not much we can do about these two weaknesses. We tried to improve the paper with respect to the second weakness mentioned by the referee, in that the paper does not properly discuss the relation with, in particular, the literature on the orbifold construction in CFT and extended chiral algebras. We added a discussion about this to the manuscript.

Below, we go through the list of requested changes in more detail.

Best regards,

Ville Lahtinen
Teresia Månsson
Eddy Ardonne

---

## Round 2 · List of Changes

Requested changes
1- As mentioned above, they need to at least briefly discuss extended chiral algebras, and the relation to the pure Virasoro characters to extended characters. They should also be precise the relation of their construction to the orbifold construction.

Answer. We agree that the paper benefits from having a discussion on extended chiral algebras and the relation with the orbifold construction. One can either do this rather extensively, or concisely. We chose the latter, and briefly discuss these matters in section 2.

2- At the beginning of section 2, they omit some key words. The partition function of every 2d CFT ON THE TORUS must be modular invariant.

Answer. This is of course true, we added this, and the implication for the current setting, namely the spectrum of periodic one dimensional chains is given in terms of a modular invariant partition function.

3- Again at the beginning of section 2, for many properties of a CFT (e.g. the specific heat), it DOES NOT MATTER IF YOU ARE ON THE TORUS, and so the distinction between different modular invariants is not important. And as I said above, the literature (especially in the '80s) is filled with CFT papers discussing other modular invariants. The authors don't mention these, and don't mention any physical properties that are different (other than the excited state spectrum).

Answer. It is of course true that for many (bulk) properties, the boundary conditions do not matter, and the referee mentions the most important one, the specific heat, which is determined by the central charge. In general, however, finite temperature properties, do depend on the details of the (at least low-energy part of the) spectrum. These properties deserve more attention in the context of quantum chains. In addition, properties related to dynamics also depend on the full spectrum. We added a short discussion on this, and added references to some '80s papers, discussing different modular invariants.

4- on p.7, they say condensation makes less fields. That's not necessarily true, since in condensations, you double other fields.

Answer. Agreed, but we don't know any example where the number of fields increases. We added the word `generically'.

5- when discussing the Z_2 x Z_2 case, they need to mention GInsparg.

Answer. We added the reference to Ginsparg.

6- they should mention that their added terms to the Hamiltonian are conventionally called "twisted" boundary conditions, and are widely studied.

Answer. Well, the terms we add do not quite correspond to conventional twisted boundary conditions. For our construction, it is really important that the twist actually depends on the symmetry sector, which is not the case in the, indeed widely studied, twisted boundary conditions. We added a sentence towards the end of sec 2.2 to discuss this.

7- at the beginning of sec 3.1, I don't know what "on-site" Z_k symmetry means. If this is on-site, what is off-site?

Answer. We dropped the "on-site" here.

8- at the beginning of sec 4.2, they mention that the Bethe ansatz solution gives a factor of 2 in the relative Fermi velocities between the two critical points in Potts. That presumably means if you take the Hamiltonian limit of the same classical 3-state Potts model, this relation holds. That's maybe interesting, but why is this relevant here? They're not studying the classical model.

Answer. Up to this point, we were taking products of the same critical points, which meant that the velocity simply amounted to an overall scale of the energies. Here, however, we combine two different critical points, each with their own velocity. In order for the construction to work, we should rescale the on of critical points, such that both have the same velocity. This is the reason for the factor of 2 in the second term of the hamiltonian in eq. (34). Without this factor of 2, the spectrum would not be described by su(2)_3, but some deformed version thereof.

9- they should mention that the "other" critical point for 3-state corresponds to the anti-ferro Potts model. The fact that this is c=1 goes back at least to Saleur, Nucl.Phys. B360 (1991) 219

Answer. We added the reference to Saleur.

10- for general k, they shouldn't say they're generalizing the Potts chain. The conventional definition of the Potts chain is that it has S_k permutation symmetry, and it is not critical for k>4. The points they describe are usually called the Z_k parafermion critical points, and this is what their modular invariants are related to.

Answer. We tried to put things in a broader perspective, but we should of course not go against conventional nomenclature, so we updated the text to reflect this.

11- in the conclusion, they mention getting SU(2)_3 by combining ferro and anti-ferro Potts chains, and speculate this may be true for other k. That seems unlikely, unless the antiferromagnetic point for Z_k parafermions is always c=1. (If that's true, the authors should note it -- I don't know offhand the answer myself).

Answer. It is indeed true that the antiferromagnetic point for the Z_k parafermions is always c=1 (Albertini). In the mean time, it was shown that for k=5,7, the critical point is u(1)_2k, as required. We included both results in the conclusions.

You are currently on this page

Resubmission 1709.04259v2 on 18 May 2021

---

## Editorial Decision

published